# Rapid sensing of hidden objects and defects using a single-pixel diffractive terahertz sensor

Jingxi Li [1,2,3], Xurong Li [1,3], Nezih T. Yardimci [1,3], Jingtian Hu[1,2,3], Yuhang Li[1,2,3], Junjie Chen[4], Yi-Chun Hung[1], Mona Jarrahi [1,3] & Aydogan Ozcan [1,2,3] ✉

Terahertz waves offer advantages for nondestructive detection of hidden objects/defects in materials, as they can penetrate most optically-opaque materials. However, existing terahertz inspection systems face throughput and accuracy restrictions due to their limited imaging speed and resolution. Furthermore, machine-vision-based systems using large-pixel-count imaging encounter bottlenecks due to their data storage, transmission and processing requirements. Here, we report a diffractive sensor that rapidly detects hidden defects/objects within a 3D sample using a single-pixel terahertz detector, eliminating sample scanning or image formation/processing. Leveraging deep-learning-optimized diffractive layers, this diffractive sensor can all-optically probe the 3D structural information of samples by outputting a spectrum, directly indicating the presence/absence of hidden structures or defects. We experimentally validated this framework using a single-pixel terahertz time-domain spectroscopy set-up and 3D-printed diffractive layers, successfully detecting unknown hidden defects inside silicon samples. This technique is valuable for applications including security screening, biomedical sensing and industrial quality control.

Inspecting hidden structures within materials or samples represents a critical requirement across various applications, such as security screening, industrial manufacturing and quality control, medicine, construction, and defense. Non-invasive detection systems based on terahertz technology offer unique opportunities for this purpose due to the ability of terahertz waves to penetrate through most optically-opaque materials and grasp the molecular fingerprint information of the sample through the rich spectral signatures of different materials in the terahertz band[1–12]. For example, terahertz time-domain spectroscopy (THz-TDS) systems have been extensively used in various non-destructive quality control applications since they can provide frequency-resolved and time-resolved responses of hidden objects[13–16]. However, existing THz-TDS systems (including reflective versions) are single-pixel and require raster scanning to acquire the image of the hidden features, resulting in relatively low-speed/low-throughput systems. Nonlinear optical processes can also be utilized to convert the terahertz information of the illuminated sample to the near-infrared regime to visualize the hidden structural information of the sample through an optical camera without raster scanning[17–20]. However, these imaging systems offer relatively low signal-to-noise ratio (SNR) levels and require bulky and expensive high-energy lasers to offer acceptable nonlinear conversion efficiencies. Alternatively, terahertz information of the illuminated sample can be encoded using spatial light modulators and the image data can be resolved using computational methods without raster scanning[21–28]. This approach, often known as terahertz computational ghost imaging, can achieve high image SNR

[1]Electrical and Computer Engineering Department, University of California, Los Angeles, CA 90095, USA. [2]Bioengineering Department, University of California, Los Angeles, CA 90095, USA. [3]California NanoSystems Institute (CNSI), University of California, Los Angeles, CA 90095, USA. [4]Physics & Astronomy Department, University of California, Los Angeles, CA 90095, USA. ✉e-mail: ozcan@ucla.edu

with a decent spatial resolution. However, the physical constraints of spatial light modulators operating at terahertz wavelengths limit the speed, and increase the size, cost, and complexity of these imaging systems. In addition to these, currently available terahertz focal-plane arrays based on field-effect transistors and microbolometers offer a limited spatial resolution and do not provide time-resolved and frequency-resolved image data, limiting the types of structural information that can be detected[29,30]. Due to these limitations, the space-bandwidth products (SBPs) of existing terahertz imaging systems are orders of magnitude lower than their counterparts operating in the visible band, thereby constraining the system's overall information throughput and its capacity to adequately capture the desired details of the hidden structures of interest.

Apart from these limitations of existing terahertz imaging systems, the identification of hidden structural features in test volumes through the processing of large-pixel-count image data is, in general, bottlenecked and challenging to reach high throughputs needed in many applications (e.g., industrial quality control and security screening) due to the digital storage, data transmission, and image processing/classification requirements that are demanding for continuous imaging and sensing systems.

Here, we present a diffractive sensor (Fig. 1) that can rapidly detect hidden defects or objects within a target sample volume using a single-pixel spectroscopic terahertz detector. Unlike traditional approaches that involve point-by-point scanning and digital reconstruction of the target sample volume using a computer, this single-pixel diffractive sensor rapidly inspects the volume of the test sample illuminated with terahertz radiation, without the formation or digital processing of an image of the sample. Stated differently, rather than formulating the detection and classification of defects or hidden objects as part of a standard machine vision pipeline (i.e., image, digitize, and then analyze using a computer), instead, we treat the detection system as a coherent diffractive optical system that processes terahertz waves on demand, which can all-optically search for and classify undesired or unexpected sources of secondary waves generated by diffraction through hidden defects or structures. In this sense, the diffractive defect sensor can be considered an all-optical sensor for unexpected or hidden sources of secondary waves within a test volume, which are detected through a single-pixel spectroscopic

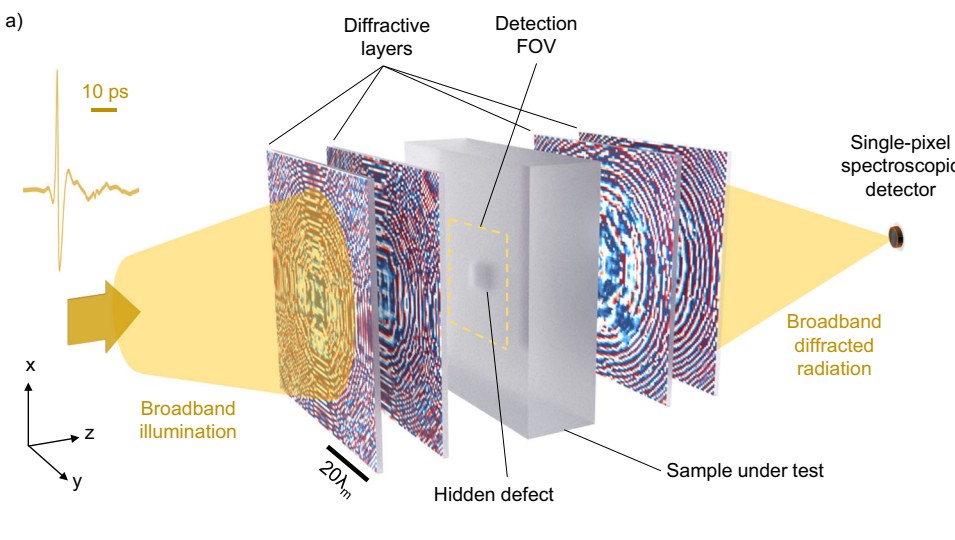

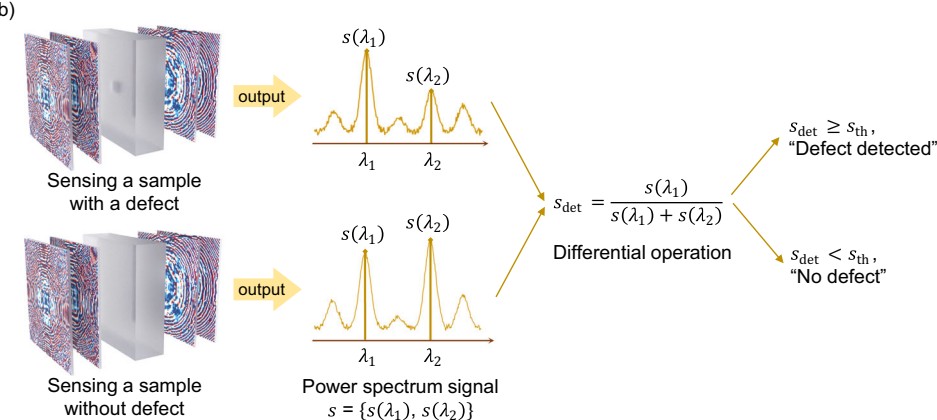

**Fig. 1 | Schematic of a diffractive terahertz sensor for rapid sensing of hidden objects or defects using a single-pixel spectroscopic detector. a,** Illustration of a single-pixel diffractive terahertz sensor. The analysis and sensing of hidden defects are performed all-optically using passive and spatially-structured diffractive layers that encode the presence or absence of unknown defects hidden within the target sample volume into the output terahertz spectrum, which is then detected using a single-pixel spectroscopic detector. The input illumination of the system shown here is provided by an ultrashort terahertz pulse, with an overall duration of ~320 ps. For illustrative purposes, only the segment with a significant magnitude is shown. **b** Working principle of the all-optical hidden object/defect detection scheme. The spectral intensity values, $s(\lambda_1)$ and $s(\lambda_2)$, sampled at two predetermined wavelengths $\lambda_1$ and $\lambda_2$ by the single-pixel detector are used to compute the output score for indicating the existence of hidden 3D defects/structures within the sample volume. $\lambda_m = (\lambda_1 + \lambda_2) / 2$.

detector. Our design is comprised of a series of diffractive layers, optimized to modify the spectrum of the terahertz radiation scattered from the test sample volume according to the absence or presence of hidden structures or defects. The diffractive layers are jointly optimized using deep learning, and contain tens of thousands of sub-wavelength phase features. Once their deep learning-based training is complete (which is a one-time effort), the resulting diffractive layers are physically fabricated using 3D printing or additive manufacturing, which forms an optical neural network[31–44]. When the test object volume is illuminated with terahertz radiation, the scattered terahertz waves from the object volume are all-optically processed by the diffractive network and sampled by a single-pixel spectroscopic detector at the output aperture of the system. The measured spectrum reveals the existence of hidden defects/structures within the sample volume all-optically, without the need for raster scanning or any image reconstruction or processing steps. Since these target structures or defects of interest are hidden within a solid volume, traditional machine vision approaches that operate at visible wavelengths cannot provide an alternative approach for these tasks. We demonstrated a proof-of-concept of this diffractive terahertz sensor by detecting hidden defects in silicon samples, which were prepared by stacking two wafers; one wafer containing etched defects and the other wafer covering the defective regions. The diffractive layers were designed to introduce a differential variation in the peak spectral intensity near two predetermined terahertz wavelengths. This diffractive defect sensor was realized using a single-pixel THz-TDS system with a plasmonic nanoantenna-based source[45,46] generating pulsed terahertz illumination and a plasmonic nanoantenna-based detector[47] sampling the terahertz spectrum at the output aperture. We numerically analyzed the performance of our diffractive defect sensor by evaluating its detection sensitivity as a function of the size and the position of the hidden defects within the detection field-of-view (FOV), also covering small feature sizes that are close to the diffraction limit of light. We fabricated the optimized diffractive layers using a 3D printer and conducted experimental tests for hidden defect detection. Our experimental results on silicon wafers with various unknown defect sizes and positions showed a good agreement with our numerical analysis, successfully revealing the presence of unknown hidden defects.

Although the diffractive defect sensors reported in this work were primarily designed for the terahertz band, the underlying concept and design approaches are also applicable for defect detection in other parts of the spectrum, including infrared, visible, and X-ray. These unique capabilities of performing computational sensing without a digital computer or the need for creating a digital 3D image will inspire the development of new task-specific all-optical detection systems and smart sensors. These systems can find diverse applications, such as industrial manufacturing and quality control, material inspection, detection/classification of hidden objects, security screening, and anti-counterfeiting measures. The non-destructive and non-invasive nature of this technology platform also makes it a valuable tool for sensitive applications, e.g., cultural heritage preservation and biomedical sensing. We believe that this framework can deliver transformative advances in various fields, where defect detection and materials diagnosis are of utmost importance.

## Results

Our reported approach demonstrates all-optical detection of hidden structures within 3D samples, enabled by a single-pixel spectroscopic terahertz detector, entirely eliminating the need to scan the samples or create, store, and digitally process their images. Our design employs an optical architecture featuring a passive diffractive encoder that generates structured illumination impinging onto the 3D sample of interest, coupled with a diffractive decoder that performs space-to-spectrum transformation, achieving defect detection based on the

optical fields scattered from the sample volume. Leveraging this synergy between the two diffractive networks and their joint training/optimization, this single-pixel defect processor offers distinct advantages compared to the existing terahertz imaging and sensing systems used for the same purpose. First, the hidden defect detection is accomplished using a single-pixel spectroscopic detector, eliminating the need for a focal plane array or raster scanning, thus greatly simplifying and accelerating the defect detection process. Second, the diffractive layers we employ are passive optical components, enabling our diffractive defect sensor to analyze the test object volume without requiring any external power source except for the terahertz illumination and single-pixel detector. Third, our all-optical end-to-end detection process negates the need for memory, data/image transmission, or digital processing using e.g., a graphics processing unit (GPU), resulting in a high-throughput defect detection scheme. Overall, these characteristics render our single-pixel diffractive terahertz sensors particularly well-suited for high-throughput screening applications such as in industrial settings, e.g., manufacturing and security. These applications require high-throughput defect detection, where the hidden defects or objects of interest are often rare, but critically important to catch. Unlike conventional imaging-based methods, which are often hindered by the 3D image data overload due to redundant information and limited frame rates of 2D image sensors, our non-imaging and single-pixel defect detection approach can deliver markedly higher sensing throughput while offering cost-effectiveness and simplicity.

### Design of the single-pixel diffractive terahertz sensor

Figure 1 illustrates the basic principles of the proof-of-concept for our single-pixel diffractive terahertz sensor design. The forward model of this design can be treated as a coherent optical system that processes spatially coherent terahertz waves at a predetermined set of 2 wavelengths ($\lambda_1$ and $\lambda_2$), where the resulting diffraction and interference processes are used for the defect detection task. As depicted in Fig. 1a, a set of diffractive layers is positioned before the sample under test to provide spatially coherent, structured broadband illumination within a given detection FOV, acting as an all-optical front-end network that is trainable. Another group of diffractive layers, positioned after the test sample, acts as the jointly trained back-end network, which all-optically performs the detection of hidden defects inside the target sample by encoding the defect information into the power spectrum at the single-pixel output aperture. This output spectrum is measured at two predetermined wavelengths, $\lambda_1$ and $\lambda_2$, producing the spectral intensity values $s(\lambda_1)$ and $s(\lambda_2)$ that yield a normalized detection score $s_{det} = \frac{s(\lambda_1)}{s(\lambda_1) + s(\lambda_2)}$ ($s_{det} \in (0,1)$); see Fig. 1b. By comparing $s_{det}$ at the single-pixel output with a pre-selected threshold $s_{th}$, the defect inference is performed to predict the existence of hidden defects within the target sample volume, i.e., $s_{det} \geq s_{th}$ for a defect, while $s_{det} < s_{th}$ indicates no defect. As will be detailed later on, we selected an unbiased threshold of $s_{th} = 0.5$ in our numerical analysis and experimental validation and, therefore a simple differential decision rule of $s(\lambda_1) \geq s(\lambda_2)$ indicates the existence of hidden defects, and $s(\lambda_1) < s(\lambda_2)$ indicates a defect-free (negative) sample. More details about the forward physical model and the joint training of the front-end and back-end diffractive layers of Fig. 1a are provided in the Methods section.

To demonstrate the feasibility of our non-destructive diffractive defect detection framework, we designed a proof-of-concept single-pixel diffractive terahertz sensor that can effectively detect pore-like hidden defects within silicon materials; these defects are not visible from the outside. Such a capability is highly sought after in numerous industrial applications due to its high prevalence and significance in determining e.g., the quality, reliability, and performance of manufactured parts/products. To achieve this capability, as depicted in

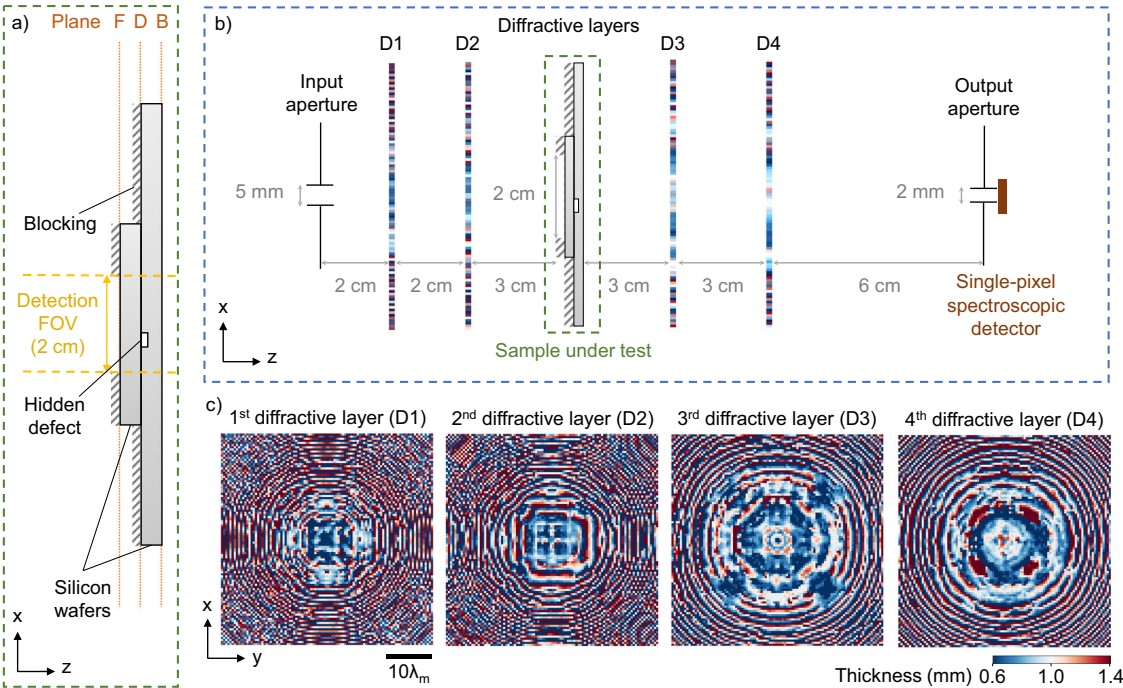

**Fig. 2 | Design of the single-pixel diffractive terahertz sensor for detecting hidden defects in silicon wafers. a** Side view schematic of the sample under test, comprising two silicon wafers stacked with a hidden defect structure fabricated on the surface of one of the wafers through photolithography and etching. The opaque regions are covered with aluminum to block the terahertz wave transmission, leaving a square-shaped opening of $2 \times 2$ cm that serves as the detection FOV. The photos showing cross-sections of a sample structure at planes F, D and B are provided in Supplementary Fig. S1. The direction of terahertz wave propagation is defined as the z direction, while the x and y directions represent the lateral directions. **b** Physical layout of the single-pixel diffractive terahertz sensor set-up, with the sizes of input/output apertures, the size of the detection FOV, and the axial distances between the adjacent components annotated. **c** Thickness profiles of the designed diffractive layers.

Fig. 2a, we created test samples with hidden defects by forming a stack of two silicon wafers that are in close contact with each other, where the surface of one of the wafers contained defect structures fabricated using photolithography and etching (see "Methods" section for details). The inspection FOV of each test object was chosen as 2×2 cm. As illustrated in Fig. 2a, an exemplary defect is located somewhere inside the detection FOV at the interface between the two wafers. Supplementary Fig. S1 includes additional photographs of a silicon test sample, showcasing its structure across 3 cross-sectional planes: planes F and B for the front and back surfaces of the stacked wafers, respectively, and plane D for the contact interface of the two silicon wafers. Such hidden defects cannot be inspected by visible or infrared cameras and would normally demand scanning imaging systems using terahertz wavelengths to form a digital image of the test object to visualize or detect potential defects using a computer.

To achieve all-optical detection of such hidden defects using our single-pixel diffractive terahertz sensor, we empirically selected $\lambda_1 = 0.8$ mm and $\lambda_2 = 1.1$ mm, and accordingly optimized the architecture of the diffractive terahertz sensor and its layers using deep learning (see the Methods section). Our single-pixel diffractive sensor design consists of four diffractive layers, with two positioned before the target sample and two positioned after the target sample, i.e., forming the front-end and back-end diffractive networks, respectively. Each of these diffractive layers is spatially coded with the same number of diffractive features (100×100), each with a lateral size of ~$0.53\lambda_m$, where $\lambda_m = (\lambda_1 + \lambda_2) / 2 = 0.95$ mm. The layout of this diffractive design is provided in Fig. 2b.

Since our diffractive sensor needs to effectively detect the hidden defects of unknown shapes and sizes that may appear *anywhere* in the target sample volume, we adopted a data-driven approach by simulating a total of 20,000 silicon test samples with hidden defects of varying sizes and shapes for training our diffractive sensor model. The

defects within these simulated test samples were set to be rectangular, with their lateral sizes ($D_x$ and $D_y$) randomly generated within a range of 1–3 mm and a depth ($D_z$) randomly chosen between 0.23 and 0.27 mm. The positions of these defects ($x_d$, $y_d$) were also randomly set within the detection FOV. We also modeled a test sample that has no defects in our numerical simulations, which forms the negative sample in our training data. To avoid our diffractive model being trained with a heavy bias towards positive samples (i.e., test samples with defects), during the formation of our training dataset, we created 20,000 replicas of our defect-free sample and mixed them with the 20,000 samples with defects, such that the final training set had a balanced ratio of positive and negative samples. We also generated a blind testing set composed of 2000 samples with various defects following the same approach; all these defective test samples were created using different combinations of parameters ($D_x$, $D_y$, $D_z$, $x_d$, $y_d$) that are uniquely different than any of those used by the training samples.

Note that our data-driven training process is a one-time effort, similar to the training effort that a digital defect analyzer based on a THz camera would need to go through (using e.g., supervised learning) in an industrial or security setting. In the training of our diffractive designs, we used a focal cross-entropy loss; see the Methods section[48]. This type of loss function can effectively reduce the penalization from samples that can be easily classified, such as those containing large hidden defects, thereby providing better detection sensitivity for more challenging samples, such as those with smaller-sized hidden defects. Moreover, in the training loss function, we also incorporated a term to impose constraints on the energy distribution of the output power spectrum (see the Methods section). This loss term aimed to maximize the output diffraction efficiency at $\lambda_1$ and $\lambda_2$, while minimizing it at other neighboring wavelengths, which helped us enhance the single-pixel SNR at the desired operational wavelengths ($\lambda_1$ and $\lambda_2$). This design choice reduced the single-pixel output at other wavelengths,

increasing our designs' experimental robustness. Additional details regarding the training data generation and the loss function that we used can be found in the Methods section.

## Numerical results and performance analysis

Figure 2c shows the resulting diffractive layers after the training was complete. We numerically tested this diffractive sensor design using the testing set containing 2,000 defective samples that we generated without any overlap with the training defective samples, as well as a test sample without any defects. By using an unbiased classification threshold of $s_{th} = 0.5$, we found that 89.62% of the defective test objects were successfully classified, and the defect-free test sample was also correctly identified. These results confirm that our diffractive model with $s_{th} = 0.5$ could achieve 100% specificity (false positive rate, FPR = 0%), while possessing a high sensitivity (i.e., true positive rate, TPR = 89.62%) to successfully detect various hidden defects with unknown combinations of shapes and locations, demonstrating its generalization for hidden defect detection. Optimization of the value of $s_{th}$ based on the training or validation sets results in $s_{th} = 0.4989$, which further improves our blind detection sensitivity (TPR) to 90.48% for the defective test samples, while maintaining a specificity of 100%. However, the value of the threshold in this case ($s_{th} = 0.4989$) is very close to the output score of a defect-free sample (i.e., $s_{det(negative)} = 0.4988 < s_{th}$), and this can potentially introduce false positives under experimental errors. Therefore, to be resilient to experimental imperfections and suppress potential false positives, we selected an unbiased threshold of $s_{th} = 0.5$, where $s(\lambda_1) \geq s(\lambda_2)$ indicates the existence of hidden defects and $s(\lambda_1) < s(\lambda_2)$ indicates a defect-free sample.

To comprehensively evaluate the efficacy of our single-pixel diffractive defect detection framework with a decision threshold of $s_{th} = 0.5$, we performed an in-depth analysis of the performance of our diffractive sensor. First, we evaluated the impact of the defect geometry on the detection performance by testing samples with rectangular defects of various dimensions ($D_x$, $D_y$, and $D_z$) located randomly within the detection FOV. For a given combination of $D_x$, $D_y$, and $D_z$, we simulated 100 test defective samples and obtained the corresponding detection sensitivity; see Fig. 3a and b. By separately scanning the values of $D_x$, $D_y$, and $D_z$, we summarized the resulting defect detection accuracies of our diffractive sensor in Fig. 3c-h. We observe that, as the lateral size ($D_x$ or $D_y$) of the hidden defect reduces, the detection sensitivity decreases considerably, regardless of $D_z$. For instance, for a relatively large defect of size $D_x = D_y = 3$ mm and $D_z = 0.3$ mm located randomly across the detection FOV, the detection sensitivity (i.e., TPR) of our diffractive sensor is 100%, while it drops to 57% when both $D_x$ and $D_y$ reduce to 0.75 mm, which is smaller than $\lambda_1$ and $\lambda_2$. Additionally, the overall detection performance also shows a degradation trend as the defect depth $D_z$ is reduced. For example, for test samples with $D_x$ and $D_y$ in the range of [2.5, 3] mm, located randomly within the detection FOV, the detection sensitivity reaches ~99.7% when $D_z$ is 0.3 mm, but drops to ~81.2% as $D_z$ reduces to 0.15 mm, which is much smaller than $\lambda_1$ and $\lambda_2$. These analyses further reveal that, for a given detection sensitivity threshold of, e.g., TPR = 75%, our diffractive sensor design can achieve accurate detection of hidden defects that are $D_x$, $D_y \geq $ ~1.25 mm (~$1.32\lambda_m$) and $D_z \geq $ ~0.21 mm (~$0.22\lambda_m$) within a FOV of 2×2 cm (~$21\lambda_m \times 21\lambda_m$). It should be noted that the smallest defect used in our analysis has a size close to the diffraction limit of light in air (~$0.5\lambda_m$). Despite the fact that the diffraction limit would be smaller in a high-refractive-index material like silicon, the medium between the sample under test and the detector is air, which sets an upper limit of 1 on the effective numerical aperture (NA) of the detection system. Such small defects in general present SNR challenges for conventional imaging systems even if a diffraction-limited image were to be formed and acquired to be digitally processed.

We further evaluated the capabilities of our diffractive terahertz sensor to detect small defects located at different positions across the detection FOV (see Fig. 3i). For this analysis, the entire detection FOV of 2×2 cm was divided into a series of concentric circles of equal radius, forming ring-like regions denoted as R1 to R6 from the center to the edges. For each one of these regions, we performed n = 100 simulations of hidden defect detection, and in each simulation, a small hidden defect of size $D_x = D_y = 0.75$ mm and $D_z = 0.18$ mm was positioned at a random location within the corresponding region; see Fig. 3j. We observed that, when the hidden defect randomly appears within R1 to R4 regions, the detection sensitivity is maintained at ~100%, but it drops to <70% when the hidden defect falls within R5 (TPR = 65%) and R6 (TPR = 20%). These results reveal an outstanding detection sensitivity in a circular region with a diameter of 1.6 cm (~$16.84\lambda_m$) at the center of the detection FOV. Stated differently, compared to utilizing the entire detection FOV of 2 × 2 cm, if we slightly reduce this inspection FOV to a diameter of 1.6 cm, the detection performance of our diffractive design can be significantly improved, allowing successful detection of even smaller hidden defects of $D_x$, $D_y > $ ~0.75 mm (~$0.79\lambda_m$) and $D_z > $ ~0.18 mm (~$0.19\lambda_m$) with a sensitivity of TPR > 79%.

## Experimental validation of the diffractive terahertz sensor

Next, we performed an experimental validation of our diffractive design using a THz-TDS set-up with a plasmonic photoconductive source and single-pixel detector; see Fig. 4a. The diffractive layers of the design were fabricated through 3D printing, and the photos of these fabricated diffractive layers are shown in Fig. 4b. After their fabrication, these layers were assembled and aligned with the test samples using a 3D-printed holder, forming a physical diffractive terahertz defect sensor that is integrated with a THz-TDS set-up; see Fig. 4c, d. Ten different exemplary silicon test samples, with or without hidden defects, were prepared for this experimental testing, where the hidden structures of these samples at plane D are shown in Fig. 5b. Among these test samples, No. 1–9 contained a hidden defect that cannot be visibly seen as the defect is located at plane D (between the two wafers); these defects possess different sizes and shapes (defined by $D_x$, $D_y$, and $D_z$). Importantly, the combinations of the parameters ($D_x$, $D_y$, $D_z$, $x_d$, $y_d$) for these test samples were never used in the training set, i.e., these 9 test samples containing defects were new to our trained diffractive model. Furthermore, the defect samples No. 8 and No. 9 had unique characteristics in their defect structures: the defect in test sample No. 8 is a rectangle of 1 × 5 mm, a shape never included in the training set; and the defect in test sample No. 9 is a 1×3 mm rectangle but rotated 45°, where such a rotation was never seen by the diffractive model in the training stage. The specific geometric parameters ($D_x$, $D_y$, $D_z$, $x_d$, $y_d$) of each test object are provided in Fig. 5e. The other test sample, i.e., sample No. 10, contains no defects, i.e., represents the negative test sample.

During our experiments, each test sample was measured 5 times, producing output power spectra shown in Fig. 5e (solid lines), which are compared to their numerically generated counterparts using the trained forward model (dashed lines). Despite the 3D fabrication errors/imperfections, possible misalignments, and other sources of error in our experimental set-up, there is a good agreement between our experimental results and the numerical predictions. However, the measured spectral curves shown in Fig. 5e exhibit minor random fluctuations and some small shifts towards longer wavelengths. To improve our SNR and build experimental resilience for hidden defect detection, we averaged the 5 spectral measurements for each test object and used the two peak spectral values at the resulting average spectrum for $s(\lambda_1)$ and $s(\lambda_2)$; see Fig. 5c. Stated differently, these 5 measurements collectively contributed to the representation of a single data point (i.e., the final detection score $\hat{s}_{exp}(\lambda)$) in Fig. 5c. Since we chose an unbiased detection threshold of $s_{th} = 0.5$, the straight line

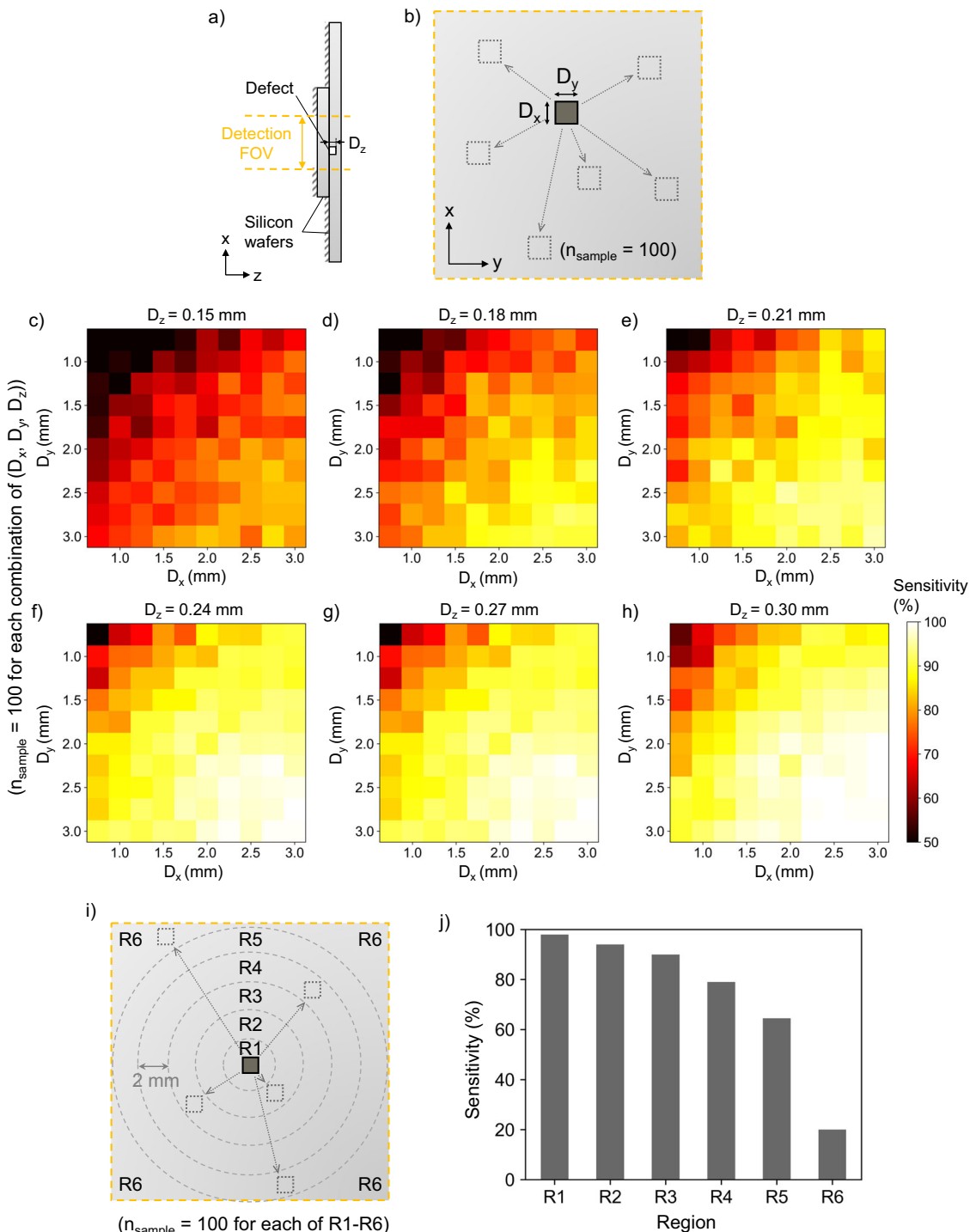

**Fig. 3 | Performance analysis of the single-pixel diffractive terahertz sensor design for detecting defects with different geometrical parameters and positions hidden inside the silicon test sample. a, b** Illustration for analyzing the impact of the shape and size of the hidden defect, defined by the lateral sizes, $D_x$ and $D_y$, and the depth, $D_z$, on the detection sensitivity of the diffractive terahertz sensor design. For a given combination of $(D_x, D_y, D_z)$, a total of $n_{sample} = 100$ test samples were numerically generated for each region (R1-R6), with each sample containing a hidden defect with a dimension of $(D_x, D_y, D_z)$ located randomly across the detection FOV. **c-h,** Defect detection accuracies as a function of the defect dimensions ($D_x$, $D_y$ and $D_z$) that are defined in **a** and **b**. **i** Illustration for analyzing the impact of the position of the hidden defect within the detection FOV on the detection sensitivity of the diffractive terahertz sensor. For this analysis, the entire detection FOV is virtually divided into a series of concentric circles of equal radius, forming ring-like regions, R1 to R6. For each of these regions, a total of $n_{sample} = 100$ test samples were numerically generated, each containing a hidden defect located randomly within the region. **j** Resulting defect detection accuracies within different regions of the detection FOV that are defined in **i** using randomly located hidden defects with $D_x = D_y = 0.75$ mm and $D_z = 0.18$ mm.

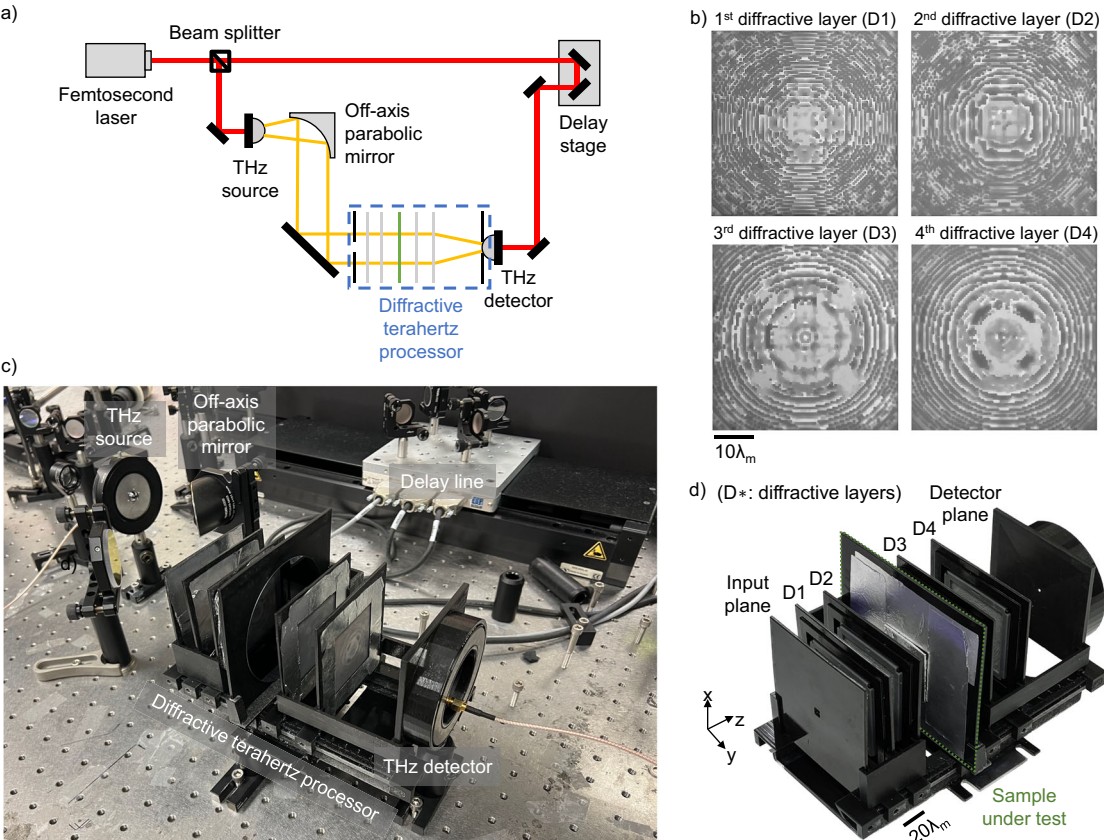

**Fig. 4 | Experimental set-up of the single-pixel diffractive terahertz sensor.**
**a** Schematic of the THz-TDS set-up. Red lines represent the propagation path of the femtosecond pulses generated from a Ti:Sapphire laser operating at 780 nm to pump/probe the terahertz source/detector. Yellow lines depict the propagation path of the terahertz pulses, which are generated with a peak frequency of ~500 GHz and a bandwidth of ~5 THz and modulated by the 3D-printed diffractive terahertz sensor to inspect the hidden structures within the test sample. **b** Photographs of the 3D-printed diffractive layers. **c** Photograph of the experimental set-up. **d** Photograph of the assembled diffractive terahertz sensor. $\lambda_m = (\lambda_1 + \lambda_2)/2 = 0.95$ mm.

of $s(\lambda_1) = s(\lambda_2)$ is used as the boundary for judging the presence or absence of defects: $s(\lambda_1) \geq s(\lambda_2)$ indicates the existence of hidden defects and $s(\lambda_1) < s(\lambda_2)$ indicates a defect-free sample. As shown in Fig. 5c, the spectral data points corresponding to test samples No. 1–9 all lie above this decision boundary, indicating that these samples were predicted by our diffractive sensor to contain hidden defects, demonstrating successful detection (true positive decisions). Meanwhile, the spectral data point for sample No. 10, which is defect-free, fell on the other side of this decision boundary, also revealing a correct inference (a true negative decision). Note that in our experiments, the smallest defects we used have a size of $1 \times 3$ mm$^2$ ($3 \times 1$ mm$^2$), and such defects in samples No. 4, 6, and 7 were all located near the edges of the detection FOV, presenting particularly challenging detection cases. This small defect area of 3 mm$^2$ approaches the smallest hidden feature that our diffractive sensor can detect as characterized in our simulation analysis (1.25 mm × 1.25 mm ≈ 1.56 mm$^2$). Therefore, these experimental results constitute compelling evidence demonstrating the practical feasibility of our diffractive defect sensor.

Next, we explored the false positive rate of our diffractive defect sensor and conducted new experiments with another defect-free (negative) test sample that was measured 10 times through repeated measurements. We formed 252 unique combinations of these 10 spectral measurements in groups of 5, and for each random combination of measurements, we averaged the 5 spectral measurements for the defect-free test object and used the peak spectral values at the resulting average spectrum for $s(\lambda_1)$ and $s(\lambda_2)$ – same as before. The results of this analysis are reported in Fig. 5d, where we observed an FPR of ~10.7% for an unbiased detection threshold of $s_{th} = 0.5$.

## Discussion

We presented an all-optical, end-to-end diffractive sensor for the rapid detection of hidden structures. This diffractive THz sensor features a distinctive architecture composed of a pair of encoder and decoder diffractive networks, each taking the unique responsibilities of structured illumination and spatial-spectral encoding, a configuration that has never been showcased in previous literature on diffractive networks. Based on this unique framework, we demonstrated a proof-of-concept hidden defect detection sensor. The success of our experimental results and analyses confirmed the feasibility of our single-pixel diffractive terahertz sensor using pulsed illumination to identify various hidden defects with unknown shapes and locations inside test sample volumes, with minimal false positives and without any image formation, acquisition, or digital processing steps.

We would like to further underscore some of the unique aspects of our reported framework over conventional approaches that utilize terahertz focal plane arrays or reflective TDS-based machine vision systems. Let us consider a common scenario in industrial quality control systems or security applications: hidden defects are discovered in only a minute fraction of the inspected objects, for example, just one in every 1000 or 10,000 objects. This means that, in more than 99.9% of the inspected cases, the objects encountered are ordinary and defect-free. However, for every inspected object, traditional terahertz imaging systems must undergo the following steps, one by one: (1) recording and reconstructing 3D images of each of these >1000–10,000 objects using a focal plane array and/or a scanning beam system, (2) sequentially storing these 3D images and uploading them to a remote or local server, and then (3) running

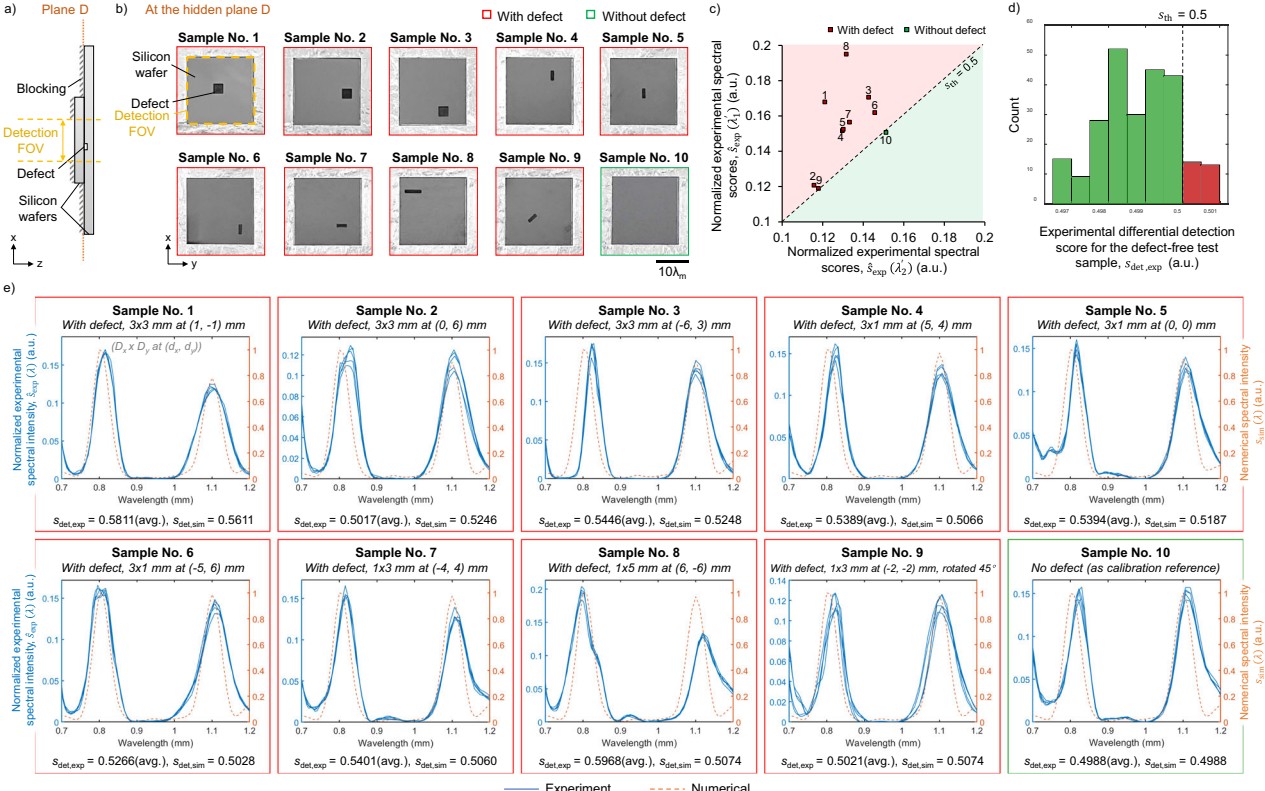

**Fig. 5 | Experimental results for detecting defects hidden inside the test samples using the single-pixel diffractive terahertz sensor. a** Illustration of the sample under test. **b** Photographs of the exemplary test samples used for the experimental blind testing, which reveal the hidden structures at the cross-sectional plane D (not visible from outside). The first nine of the test samples (i.e., samples No. 1–9) contain etched defects that have different shapes and are positioned at different locations within the detection FOV, while the last sample (i.e., sample No. 10) has no defects. These photos were captured by removing the smaller silicon wafer at the front, i.e., the left wafer in **a. c** Normalized experimental spectral scores for the test samples shown in **b. d** Histogram showing the distribution of the 252 experimental differential detection scores, which were obtained through measuring a defect-free (negative) sample 10 times through

repeated experiments and combining these 10 spectral measurements in unique groups of 5, each resulting in an experimental differential detection score, $s_{\mathrm{det,exp}}$, based on the average spectrum. Note that $C(^{10}_5) = 252$, where $C$ refers to the combination operation. **e** Normalized experimental spectral intensity (solid lines) for the different test samples shown in **b**, compared with their numerically simulated counterparts (dashed lines). Each test sample was measured 5 times and the results of all 5 measurements are shown in the same graph. For the test samples with defects (i.e., samples No. 1–9), the lateral sizes ($D_x$ and $D_y$) and positions ($x_d$, $y_d$) of the defects are shown in italicized texts, and $D_z = 0.25$ mm. The experimental differential detection scores, $s_{\mathrm{det,exp}}$, are calculated for each test sample based on averaging the 5 spectral measurements, which are also compared with their numerical counterpart $s_{\mathrm{det,sim}}$, reported at the bottom of each panel.

machine learning algorithms using GPU clusters to digitally determine the presence of hidden defects. Consequently, a substantial amount of imaging and computational resources are wasted in extracting and processing redundant information, leading to lowered detection throughput and a massive burden on data/storage and bandwidth. Even for large defects that are relatively simple to detect using state-of-the-art algorithms frequently used in machine vision, the throughput of such imaging-based solutions would still be limited by the low frame rate of the 2D image sensor-arrays; this image capture process will be even slower for various point-scanning-based THz imaging systems. In contrast, our method effectively circumvents all these steps. It does not require reconstructing or creating 3D images for each object, nor does it need to store and upload any images to the cloud, and it also certainly does not necessitate using GPU clusters to handle large volumes of data. Using a single-pixel detector and snap-shot dual illumination wavelengths for defect sensing, our method signifies a unique paradigm, eliminating the image capture/reconstruction, storage, and transmission steps needed by GPU-based digital processing systems. As a result, the volumetric defect detection rate can be elevated to align with the exceptional speed of single-pixel sensors, which can have a response time of <1 μs[49]. Therefore, with our presented framework the efficiency of terahertz defect inspection/detection can be dramatically enhanced, and concurrently the cost per inspection

can be substantially reduced using the presented diffractive defect sensors.

In our experimental results reported earlier using our single-pixel diffractive defect sensor, we adopted a strategy of taking the average of $N_{\mathrm{avg}} = 5$ measurements to improve the detection SNR and the defect detection performance of the system. To delve deeper into this averaging strategy, similar to Fig. 5d, from the 10 spectral measurements of this defect-free sample, we randomly selected $N_{\mathrm{avg}}$ samples to form a group for averaging, and accordingly took the spectral peaks in the resulting spectrum as $s(\lambda_1)$ and $s(\lambda_2)$ to obtain the detection score $s_{\mathrm{det}}$. After analyzing all the detection results corresponding to all the possible combinations, i.e., $C(^{10}_{N_{\mathrm{avg}}})$, we calculated the false positive rate, FPR, as a function of the averaging factor $N_{\mathrm{avg}}$. Figure 6 reports the impact of $N_{\mathrm{avg}}$ on the FPR in our detection results. These findings indicate that when $N_{\mathrm{avg}}$ is 1, i.e., no averaging is used, the FPR reaches as high as ~30%. However, as $N_{\mathrm{avg}}$ increases, the reported FPR begins to decrease significantly. Specifically, when $N_{\mathrm{avg}} = 5$, the FPR falls to 10.7%, which aligns with our earlier results in Fig. 5d. When $N_{\mathrm{avg}}$ further increases to 8, the FPR drops to 0%, indicating that false positives can be eliminated by this averaging strategy (see Fig. 6).

Another aspect we would like to discuss pertains to the variation of our diffractive defect sensor's performance at different sub-regions within the detection FOV. For this, we simulated the terahertz wave

field $E_{detFOV}$ within the detection FOV at plane D, i.e., the plane where the incoming terahertz field interacts with the potential hidden defects (Supplementary Fig. 2a). The amplitude and phase distributions of $E_{detFOV}$ at the two predetermined wavelengths ($\lambda_1 = 0.8$ mm and $\lambda_2 = 1.1$ mm) are shown in Supplementary Fig. 2b–e. Interestingly, $E_{detFOV}(\lambda_1)$ exhibits a relatively uniform amplitude and phase distributions at the center, with the phase profile beginning to present faster changes as moving away from the center. On the other hand, the amplitude and phase distributions present a specific structure, featuring higher intensity in the center and lower intensity around. These observations suggest that the terahertz light reaching the detection FOV, after being processed by the encoding diffractive network, primarily forms structured light illumination at $\lambda_2$, while it largely maintains a spherical wave-like illumination pattern at $\lambda_1$. We further analyzed the influence of this structured light illumination on the defect detection results. As illustrated in Supplementary Fig. 2f, we assumed a hidden defect with $D_x = D_y = 3$ mm, $D_z = 0.25$ mm that can be located *anywhere* within the detection FOV, identical to the one in Sample No. 1 used in our experimental validation. We numerically quantified the output spectral intensity at the two operational wavelengths, $s(\lambda_1)$ and $s(\lambda_2)$, as a function of the hidden defect's position within the detection FOV, and calculated the resulting final detection score $s_{det}$ in each case. These results are summarized in Supplementary Fig. 2g–i. As shown in Supplementary Fig. 2j, the detection scores $s_{det}$ within the detection FOV exceed the detection threshold $s_{th} = 0.5$, indicating 100% true positive detection, except at the edges of the detection FOV, which have relatively poor representation during the training phase. It can be seen that, $s(\lambda_1)$ presents a fairly uniform distribution (excluding the edges); in contrast, the overall distribution of $s(\lambda_2)$ shows a significant negative correlation with $s_{det}$, and they both exhibit substantial correspondence with the amplitude distribution of $E_{detFOV}(\lambda_2)$ as shown in Supplementary Fig. 2d. This finding suggests that $E_{detFOV}(\lambda_1)$ and $E_{detFOV}(\lambda_2)$ essentially play roles akin to reference light and probe light, respectively. Here, the intensity structure of this "probe" light $E_{detFOV}(\lambda_2)$ notably contributes to the distribution of $s_{det}$, which both exhibit features of higher values at the center and lower values at the periphery, correlating with our observations in Fig. 3j.

Different approaches can be explored to further enhance the performance of our defect detection sensor. To achieve a larger detection FOV, one will need to enlarge the diffractive layers so that the possible defect information can be effectively processed by the diffractive layers using a larger NA. Additionally, the number of trainable diffractive layers in both the front-end and back-end optical networks can be increased to improve the approximation power of the diffractive network by creating a deeper diffractive sensor[40,50]. Furthermore, if the detection of defects with even smaller features is required, we can use shorter terahertz wavelengths, with accordingly smaller diffractive features fabricated as part of each diffractive layer.

In the demonstrated diffractive defect sensor, we utilized only two wavelength components at the output power spectra of the single-pixel detector to encode the defect information, and did not leverage the entire spectral bandwidth provided by the THz-TDS system. Our decision to use the THz-TDS system was primarily driven by the availability of hardware resources in our laboratory. Therefore, by using a THz illumination set-up with two pre-determined wavelengths ($\lambda_1$ and $\lambda_2$), the complexity of our current system would be significantly simplified, with costs substantially reduced. Furthermore, it is also conceivable to train a diffractive defect sensor that utilizes more wavelengths, encoding additional information regarding the hidden defects such as e.g., the size and material type of the defect, which may lead to more comprehensive defect detection and classification capabilities. To achieve spectral encoding of such defect information using more wavelengths, a larger number of trainable diffractive features per design would, in general, be required; this increase would be

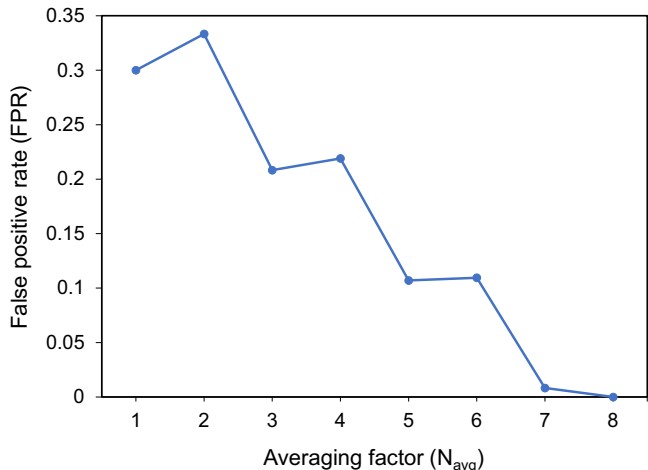

**Fig. 6 | The impact of different averaging factors ($N_{avg}$) used for experimental spectral measurements on the false positive rate (FPR) in our defect detection results.** For this analysis, a defect-free (negative) sample was measured 10 times through repeated experiments; using $C\left(0ex10N_{avg}\right)$, we selected within these 10 spectral measurements all the possible combinations of $N_{avg}$, each resulting in an experimental differential detection score, $s_{det,exp}$, based on the average spectrum. By comparing these $s_{det,exp}$ with the detection threshold $s_{th} = 0.5$, the FPR as a function of $N_{avg}$ was obtained.

approximately proportional to the number of wavelengths used to encode independent channels of information[44].

While the presented single-pixel terahertz sensor enabled high-throughput detection of defects with feature sizes close to the diffraction limit of light, the maximum thickness of the test sample that can be probed in transmission mode would be limited by the terahertz absorption or scattering inside the sample volume. For highly absorbing samples or samples with metal cores, for example, the proposed transmission system will present limitations to probe deeper into the test sample volume. However, this limitation is not specific to our diffractive sensor design, and is in fact commonly shared by all terahertz-based imaging and sensing systems. In case the terahertz transmission from certain test samples creates major SNR challenges due to terahertz absorption and/or scattering within the depths of the test sample, the presented diffractive defect sensor designs can be modified to work in reflection mode so that a partial volume of the highly absorbing and/or reflecting test objects can be rapidly probed and analyzed by our single-pixel diffractive sensor. In this reflection mode of operation, the whole deep learning-based training strategy outlined in this work will remain the same, except that between the test sample and the encoder diffractive network there will be a beam splitter (e.g., a mylar film) that communicates with an orthogonally placed diffractive decoder that will be jointly trained with the front-end diffractive encoder, following the architecture we reported earlier in our Results section. Through this reflection mode of the diffractive sensor, one can extend the applications of our all-optical hidden defect sensor to partially probe and analyze highly absorbing and/or scattering test objects that would otherwise not transmit sufficient terahertz radiation. Compared to conventional reflective THz-TDS defect detection systems used for similar purposes, which necessitate mechanical scanning to detect defects at various locations, this reflective diffractive sensor design would allow for all-optical, rapid defect detection within a large sample volume, eliminating the need for mechanical scanning or extensive data storage, transmission and digital processing.

One additional potential limitation of our framework is uncontrolled mechanical misalignments among the diffractive layers that constitute the diffractive encoder and decoder networks, as well as

possible lateral/axial misalignments that might be observed between the diffractive layers and the test sample volume. As a mitigation strategy, diffractive designs can be "vaccinated" to such variations by modeling these random variations and misalignments during the optimization phase of the diffractive sensor to build misalignment-resilient physical systems. It has been shown in our earlier works that the evolution of diffractive surfaces during the deep learning-based training of a diffractive network can be regularized and guided toward diffractive solutions that can maintain their inference accuracy despite mechanical misalignments[35,38,43]. This misalignment-tolerant diffractive network training strategy models the layer-to-layer misalignments, e.g., translations in x, y, and z, over random variables and introduces these errors as part of the forward optical model, inducing "vaccination" against such inaccuracies and/or mechanical variations. In addition to mechanical misalignments, the same training scheme can also be extended to mitigate the effects of other potential error sources e.g., fabrication inaccuracies, refractive index measurement errors, and detection noise, improving the robustness of single-pixel defect detector devices.

Finally, our framework can potentially sense the presence of even smaller hidden defects or objects with subwavelength dimensions. While our diffractive defect sensor is diffraction-limited, isolated subwavelength features/defects can still generate traveling waves (through scattering) to be sensed by our diffractive layers. This, however, does not mean our diffractive processor can resolve two closely positioned subwavelength defects or morphologically distinguish them from larger defects since it can only process propagating waves from the defect volume, without access to the evanescent waves in the near-field of a defect that carry the super-resolution information. In this respect, similar to localization-based microscopic imaging approaches, it is potentially feasible to sense isolated subwavelength defects within the sample volume without any structural fine details or the capability to distinguish them from larger defects. This would be possible only with sufficient detection sensitivity: if the weak scattering of such small defects, coupled into propagating secondary waves, can be detected within the SNR of the single-pixel defect detection system. Moreover, the presented design and the underlying concept can also be applied to other frequency bands of the electromagnetic spectrum, such as the infra-red[51-53] and X-ray[54-57], for all-optical detection of hidden objects or defects. With its unique capabilities to rapidly and accurately inspect hidden objects and identify unknown defects hidden within materials, we believe that the presented single-pixel diffractive terahertz sensor can be useful for a variety of applications, including industrial quality control, material inspection and security screening.

## Methods

### Numerical forward model of a single-pixel diffractive terahertz sensor

Our system consists of successive diffractive layers that are modeled as thin dielectric optical modulation elements of different thicknesses, where the $i^{th}$ feature on the $k^{th}$ layer at a spatial location $(x_i, y_i, z_i)$ represents a complex-valued transmission coefficient, $t^k$, which depends on the illumination wavelength ($\lambda$):

$$t^k(x_i, y_i, z_k, \lambda) = a^k(x_i, y_i, z_k, \lambda) \exp\left( j\phi^k(x_i, y_i, z_k, \lambda) \right) \quad (1)$$

where $a$ and $\phi$ denote the amplitude and phase coefficients, respectively. The diffractive layers are connected to each other by free-space propagation, which is modeled through the Rayleigh-Sommerfeld diffraction equation[31,34], with an impulse response of $f_i^k(x, y, z, \lambda)$:

$$f_i^k(x, y, z, \lambda) = \frac{z - z_k}{r^2} \left( \frac{1}{2\pi r} + \frac{1}{j\lambda} \right) \exp\left( \frac{j2\pi r}{\lambda} \right) \quad (2)$$

where $r = \sqrt{(x - x_i)^2 + (y - y_i)^2 + (z - z_k)^2}$ and $j = \sqrt{-1}$. $f_i^k(x, y, z, \lambda)$ represents the complex-valued field at a spatial location $(x, y, z)$ at a wavelength of $\lambda$, which can be viewed as a secondary wave generated from the source at $(x_i, y_i, z_k)$. Following the Huygens principle, each diffractive feature in a diffractive network can be modeled as the source of a secondary wave, as in Eq. (2). As a result, the optical field modulated by the $i^{th}$ diffractive feature of the $k^{th}$ layer ($k \geq 1$, treating the input object plane as the $0^{th}$ layer), $E^k(x_i, y_i, z_k, \lambda)$, can be written as:

$$E^k(x_i, y_i, z_k, \lambda) = t^k(x_i, y_i, z_k, \lambda) \cdot \sum_{m \in M} E^{k-1}(x_m, y_m, z_{k-1}, \lambda) \cdot f_m^{k-1}(x_i, y_i, z_k, \lambda)$$

$$(3)$$

where $M$ and $z_*$ denote the number of diffractive features on the $(k-1)^{th}$ diffractive layer and the $z$ location of the $*^{th}$ layer, respectively. The axial distances between the input/output aperture, diffractive layers and the object under test can be found in Fig. 2b.

The amplitude and phase components of the complex transmittance of the $i^{th}$ feature of diffractive layer $k$, i.e., $a^k(x_i, y_i, z_k, \lambda)$ and $\phi^k(x_i, y_i, z_k, \lambda)$ in Eq. (1), are defined as a function of the material thickness, $h_i^k$, as follows:

$$a^k(x_i, y_i, z_k, \lambda) = \exp\left( -\frac{2\pi \kappa_d(\lambda) h_i^k}{\lambda} \right) \quad (4)$$

$$\phi^k(x_i, y_i, z_k, \lambda) = (n_d(\lambda) - n_{air}) \frac{2\pi h_i^k}{\lambda} \quad (5)$$

where the wavelength-dependent dispersion parameters $n_d(\lambda)$ and $\kappa_d(\lambda)$ are the refractive index and the extinction coefficient of the diffractive layer material corresponding to the real and imaginary parts of the complex-valued refractive index $\tilde{n}_d(\lambda)$, i.e., $\tilde{n}_d(\lambda) = n_d(\lambda) + j\kappa_d(\lambda)$. These dispersion parameters for the 3D-printing material used in this work were experimentally measured over a broad spectral range (see Supplementary Fig. S3). The thickness values of the diffractive features $h_i^k$ represent the learnable parameters of our diffractive sensor devices, which are composed of two parts, $h_{trainable}$ and $h_{base}$:

$$h = h_{trainable} + h_{base} \quad (6)$$

where $h_{trainable}$ denotes the learnable thickness parameters of each diffractive feature and is confined between 0 and $h_{max} = 0.8$ mm. The additional base thickness, $h_{base}$, is a constant, which is chosen as 0.6 mm to serve as the substrate support for the diffractive layers. To achieve the constraint applied to $h_{trainable}$, an associated latent trainable variable $h_v$ was defined using:

$$h_{trainable} = \frac{h_{max}}{2} \cdot \left( \sin(h_v) + 1 \right) \quad (7)$$

Note that before the training starts, $h_v$ values of all the diffractive features were initialized as 0.

We calculated the propagation of the optical field inside the sample volume using:

$$\tilde{f}^k(x, y, z, \lambda) = \frac{z - z_s}{r^2} \left( \frac{1}{2\pi r} + \frac{1}{j\lambda} \right) \exp\left( -\frac{2\pi r}{\lambda} \left( \kappa_{object}(\lambda) - j n_{object}(\lambda) \right) \right)$$

$$(8)$$

where the wavelength-dependent parameters $n_{object}(\lambda)$ and $\kappa_{object}(\lambda)$ are the refractive index and the extinction coefficient of the object material. In this paper, since silicon wafers were used as the test

objects, $n_{object}(\lambda)$ and $\kappa_{object}(\lambda)$ were set to 3.4174 and 0, respectively (disregarding the negligible material absorption of silicon at our wavelengths of interest). After calculating the impulse response of optical field propagation inside the object volume $\tilde{f}^k$, the resulting complex field is calculated using Eq. (3), except that the $f^k$ is replaced by $\tilde{f}^k$ used for the object material. When the propagating optical field encounters a defect inside the object, we model the defect as a tiny volume element with a certain size, which can also be considered as a thin optical modulation element. Due to the difference in the material properties between the defect and the object, the presence of the defect introduces additional amplitude and phase changes, which can be calculated in our forward model using the following formula:

$$a_{defect}(x_i, y_i, z_k, \lambda) = \exp\left(-\frac{2\pi h_{defect}}{\lambda}\left(\kappa_{defect}(\lambda) - \kappa_{object}(\lambda)\right)\right) \quad (9)$$

$$\phi_{defect}(x_i, y_i, z_k, \lambda) = \left(n_{defect}(\lambda) - n_{object}\right)\frac{2\pi h_{defect}}{\lambda} \quad (10)$$

where $n_{defect}(\lambda)$ and $\kappa_{defect}(\lambda)$ are the refractive index and the extinction coefficient of the object material, $h_{defect}$ is the depth of the defect along the axial direction. In our experimental validation, these defects were fabricated by etching silicon wafers. Therefore, without loss of generality, air constitutes the material of the defects, i.e., $n_{defect}(\lambda) = n_{air}(\lambda) = 1$ and $\kappa_{defect}(\lambda) = \kappa_{air}(\lambda) = 0$.

### Spectral scores for the detection of hidden objects and defects

Assuming that the diffractive design is composed of $K$ layers (excluding the input, sample, and output planes), a single-pixel spectroscopic detector positioned at the output plane (denoted as the $(K+1)^{th}$ layer) measures the power spectrum of the resulting optical field within the active area of the detector $D$, where the resulting spectral signal can be denoted as $s(\lambda)$:

$$s(\lambda) = \sum_{(x,y)\in D}\left|E^{K+1}(x, y, z_{K+1}, \lambda)\right|^2 \quad (11)$$

For the diffractive sensor designs reported in this paper, the sizes of the detector active area and the output aperture are both set to be 2×2 mm. We sampled the spectral intensity of the diffractive network output at a pair of wavelengths $\lambda_1$ and $\lambda_2$, resulting in spectral intensity values $s(\lambda_1)$ and $s(\lambda_2)$. The output detection score $s_{det}$ of the diffractive sensor is given by:

$$s_{det} = \frac{s(\lambda_1)}{s(\lambda_1) + s(\lambda_2)} \quad (12)$$

For an unbiased detection threshold of $s_{th} = 0.5$, this Eq. (12) boils down to a differential detection scheme where $s(\lambda_1) \geq s(\lambda_2)$ indicates the existence of hidden defects and $s(\lambda_1) < s(\lambda_2)$ indicates a defect-free sample. In this paper, $\lambda_1$ and $\lambda_2$ were empirically selected as 0.8 and 1.1 mm, respectively. To analyze the classification results produced by our diffractive sensor design, we analyzed the numbers of true positive, false positive, true negative and false negative samples, denoted as $n_{TP}$, $n_{FP}$, $n_{TN}$ and $n_{FN}$, respectively. Based on these, we reported the sensitivity (i.e., true positive rate, TPR), the specificity, the false negative rate (FNR) and the false positive rate (FPR) using:

$$\text{Sensitivity} = \text{TPR} = \frac{n_{TP}}{n_{TP} + n_{FN}} = 1 - \text{FNR} \quad (13)$$

$$\text{Specificity} = \frac{n_{TN}}{n_{TN} + n_{FP}} \quad (14)$$

$$\text{False positive rate (FPR)} = 1 - \text{Specificity} = \frac{n_{FP}}{n_{TN} + n_{FP}} \quad (15)$$

During the training of our diffractive terahertz sensor model, we assumed that the optical field at the input aperture of the system has a flat spectral magnitude, i.e., the total power of the illumination beam at $\lambda_1$ and $\lambda_2$ is equal in our numerical simulations. However, the pulsed terahertz source employed in our experimental TDS set-up contained a different spectral profile within the band of operation. To calibrate our diffractive sensor system, we performed a normalization step for the experimentally measured output power spectra for all the test samples using a linear correction factor $\sigma(\lambda)$, which was obtained using:

$$\sigma(\lambda) = \frac{\frac{s_{sim}^{(r)}(\lambda_2)}{s_{exp}^{(r)}(\lambda_2)} - \frac{s_{sim}^{(r)}(\lambda_1)}{s_{exp}^{(r)}(\lambda_1)}}{\lambda_2' - \lambda_1'}(\lambda - \lambda_1') + \frac{s_{sim}^{(r)}(\lambda_1)}{s_{exp}^{(r)}(\lambda_1')} \quad (16)$$

where $s_{exp}^{(r)}(\lambda_1')$ and $s_{exp}^{(r)}(\lambda_2')$ represent the experimentally measured output spectral intensity values corresponding to the spectral peaks closest to $\lambda_1$ and $\lambda_2$, respectively, using the defect-free sample as the test object after averaging multiple spectral measurements. $s_{sim}^{(r)}(\lambda_1)$ and $s_{sim}^{(r)}(\lambda_2)$ are the numerically computed counterparts of $s_{exp}^{(r)}(\lambda_1)$ and $s_{exp}^{(r)}(\lambda_2)$, respectively. Based on $\sigma(\lambda)$, we normalized the experimental spectral curves $s_{exp}(\lambda)$ as:

$$\hat{s}_{exp}(\lambda) = \sigma(\lambda)s_{exp}(\lambda) \quad (17)$$

By following this calibration/normalization routine outlined above, our diffractive model can be used under different forms of input broadband radiation, without overfitting to any experimental radiation source, which forms an important practical advantage of our framework. Moreover, even if the experimental system contains certain manufacturing errors and misalignments that result in a mismatch with the ideal forward physical model, the spectral intensity peaks $s_{exp}^{(r)}(\lambda_1')$ and $s_{exp}^{(r)}(\lambda_2')$ near the two wavelengths $\lambda_1$ and $\lambda_2$ can still be utilized as references for the calibration of the system. Figure 5e illustrates the normalized experimental spectral intensity $\hat{s}_{exp}(\lambda)$ defined by Eq. (17) for different test samples, and their normalized peak spectral intensity values near $\lambda_1$ and $\lambda_2$, i.e., $\hat{s}_{exp}(\lambda_1')$ and $\hat{s}_{exp}(\lambda_2')$, are reflected in Fig. 5c. The experimental differential spectral scores presented in Fig. 5d were computed based on Eq. (12) by replacing $s(\lambda_1)$ and $s(\lambda_2)$ with $\hat{s}_{exp}(\lambda_1')$ and $\hat{s}_{exp}(\lambda_2')$, respectively.

### Training loss functions

The loss function used for training our presented diffractive terahertz sensor is defined as:

$$\mathcal{L}_{total} = \mathcal{L}_{det} + \alpha_{eff} \cdot \mathcal{L}_{eff} + \alpha_{ed} \cdot \mathcal{L}_{ed} \quad (18)$$

The first loss term, $\mathcal{L}_{det}$, stands for defect detection loss. To train our diffractive model to better classify challenging samples, e.g., those with small defects, we employed the focal loss[48] given by:

$$\mathcal{L}_{det} = \begin{cases} -\beta(1 - s_{det})^\gamma \log(s_{det}), & s_{GT} = 1 \\ -(1 - \beta)s_{det}^\gamma \log(1 - s_{det}), & s_{GT} = 0 \end{cases} \quad (19)$$

where $s_{GT}$ denotes the ground-truth label of the given 3D object volume, indicating the existence of the hidden defect ($s_{GT} = 1$) or not ($s_{GT} = 0$). $\beta$ denotes the coefficient to balance the loss magnitude for the positive and negative samples, and $\gamma$ is the focusing parameter used to down-weigh the importance of the easy-to-classify samples (e.g., 3D objects with relatively larger hidden defect(s)). $\beta$ and $\gamma$ were empirically chosen as 0.5 and 4 throughout our training process.

In addition to being sensitive and specific to unknown/random hidden defects, we also wanted to ensure that the single-pixel diffractive sensor is photon efficient, achieving a decent SNR at its output. Therefore, in the training loss function, we also added a loss term, $\mathscr{L}_{\text{eff}}$, to increase the diffraction power efficiency at the output single-pixel aperture. $\mathscr{L}_{\text{eff}}$ is defined as:

$$\mathscr{L}_{\text{eff}} = \begin{cases} \eta_{\text{th}} - \eta_w, & \text{if } \eta_{\text{th}} \geq \eta \\ 0, & \text{if } \eta_{\text{th}} < \eta \end{cases} \tag{20}$$

where $\eta$ denotes the diffraction power efficiency at the output detector and $\eta_{\text{th}}$ refers to the penalization threshold for $\eta$, which was empirically selected as 0.01 during the training process. $\eta$ is defined as:

$$\eta = \frac{I_{\text{detector}}}{I_{\text{input}}} \tag{21}$$

where $I_{\text{detector}}$ represents the total power of the optical field calculated within the active area $D$ of the output single-pixel detector aperture across the two wavelengths $\lambda_1$ and $\lambda_2$, i.e., $I_{\text{detector}} = \sum_{\lambda \in \{\lambda_1, \lambda_2\}} \sum_{(x,y) \in D} |E^{K+1}(x,y,z_{K+1},\lambda)|^2$, and $I_{\text{input}}$ represents the total power of the optical field right after the input aperture $P$ across the two wavelengths $\lambda_1$ and $\lambda_2$, i.e., $I_{\text{input}} = \sum_{\lambda \in \{\lambda_1, \lambda_2\}} \sum_{(x,y) \in P} |E^0(x,y,z_0,\lambda)|^2$. For the diffractive sensor design in this paper, the size of the input aperture $P$ is set as $5 \times 5$ mm.

We also incorporated another loss term, $\mathscr{L}_{\text{ed}}$, in Eq. (18) to constrain the energy distribution of the output power spectrum, such that the spectral power is maximized at the two predetermined wavelengths used for all-optical inference of defects ($\lambda_1$ and $\lambda_2$) while maintaining a relatively low power level at the other neighboring wavelengths. To achieve this, we used:

$$\mathscr{L}_{\text{ed}} = \frac{1}{4} \sum_{\lambda \in \Lambda} \sum_{(x,y) \in D} \left| E^{K+1}(x,y,z_{K+1},\lambda) \right|^2 \tag{22}$$

where $\Lambda = \{0.75, 0.85, 1.05, 1.15\}$ mm.

The hyperparameters, $\alpha_{\text{eff}}$ and $\alpha_{\text{ed}}$, in Eq. (18) are the weight coefficients associated with the diffraction efficiency penalty loss term and the spectral energy distribution loss term, respectively, which were empirically selected as 0.01 and 0.1 throughout the training process.

### Training details of the single-pixel diffractive terahertz sensor

For the numerical model used in this manuscript, the smallest sampling period for simulating the complex optical fields is set to 0.25 mm ($\sim 0.26\lambda_{\text{m}}$), which is half the lateral size of the diffractive features (i.e., 0.5 mm = $\sim 0.53\lambda_{\text{m}}$). To build a dataset for training and evaluating the diffractive sensor model, we generated a total of 24,000 defect samples with varying sizes ($D_x$, $D_y$, $D_z$) and positions ($x_d$, $y_d$), which were then divided into three sets: training, validation, and testing, each containing 20,000, 2000, and 2000 samples, respectively. We also generated a defect-free sample and mixed the replicas of that with each of these defect sample subsets at a 1:1 ratio during the training process, so that the number of samples with and without defects could be balanced. We also considered the error in the thickness of the silicon wafer in our training forward model. To mitigate its impact on the detection performance of our diffractive sensor, we modeled the silicon thickness value as a random variable that follows a uniform distribution of [0.52, 0.53] mm in our forward model. This ensures that the trained diffractive sensor can provide detection performance that is resilient to the variations in the thickness values of the silicon wafer test samples.

The single-pixel diffractive sensor model used in this work was trained using TensorFlow (v2.5.0, Google LLC). We selected Adam optimizer[58], and its parameters were taken as the default values. The batch size was set as 32. The learning rate was set as 0.001. For the training of our diffractive model, we used a workstation with a GeForce GTX 1080Ti GPU (Nvidia Inc.) and Core i7 8700 central processing unit (CPU, Intel Inc.) and 64 GB of RAM, running Windows 10 operating system (Microsoft Inc.). The training of the diffractive model was performed with 200 epochs, which typically required ~10 h. The best model was selected based on the detection performance quantified using the validation data set.

### Fabrication of the test samples

The defects on the silicon wafers were fabricated through the following procedure. First, a $SiO_2$ layer was deposited on the silicon wafers using low-pressure chemical vapor deposition (LPCVD). Defect patterns were defined by photolithography, and the $SiO_2$ layer was etched in the defect regions using reactive-ion etching (RIE). After removing the remaining photoresist, defects in silicon wafers were formed through deep reactive-ion etching (DRIE) with the $SiO_2$ layer serving as the etch mask. Finally, the $SiO_2$ layer was removed through wet etching using a buffered oxide etchant (BOE). We measured the depth of the defect regions, $D_z$, with a Dektak 6 M profilometer, which was ~0.25 mm for all the prepared test samples.

The diffractive layers were fabricated using a 3D printer (Form 3, Formlab). To assemble the printed diffractive layers and input objects, we employed a 3D-printed holder (Objet30 Pro, Stratasys) that was designed to ensure the proper placement of these components according to our numerical design.

### Terahertz time-domain spectroscopy set-up

A Ti:Sapphire laser was used to generate femtosecond optical pulses with a 78 MHz repetition rate at a center wavelength of 780 nm. The laser beam was split into two parts: one part was used to pump the terahertz source, a plasmonic photoconductive nano-antenna array, and the other part was used to probe the terahertz detector, a plasmonic photoconductive nano-antenna array providing a high sensitivity and broad detection bandwidth. The terahertz radiation generated by the source was collimated and directed to the scanned sample using an off-axis parabolic mirror, as shown in Fig. 4a. The output signal from the terahertz detector was amplified with a transimpedance amplifier (Femto DHPCA-100) and detected with a lock-in amplifier (Zurich Instruments MFLI). By changing the temporal delay between the terahertz radiation and the laser probe beam incident on the terahertz detector, the time-domain signal was obtained. The corresponding spectrum was calculated by taking the Fourier transform of the time-domain signal, resulting in an SNR of 90 dB and an observable bandwidth of 5 THz for a time-domain signal span of 320 ps.

### Reporting summary

Further information on research design is available in the Nature Portfolio Reporting Summary linked to this article.

## Data availability

All the data and methods needed to evaluate the conclusions of this work are presented in the main text and Supplementary Information. Additional data can be requested from the corresponding author.

## Code availability

The codes used in this work use standard libraries and scripts that are publicly available in TensorFlow.

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

## Acknowledgements

The Ozcan Research Group at UCLA acknowledges the support of ONR (Grant # N00014-22-1-2016). Jarrahi's group at UCLA acknowledges the support of the Department of Energy (grant # DE-SC0016925).

## Author contributions

A.O. conceived and initiated the research, J.L., X.L., N.T.Y., and Y.L. conducted experiments. X.L. prepared the test samples. J.L., X.L., and N.T.Y. processed the resulting data. J.C., J.H., and Y.H. contributed to the holder design of the diffractive sensor. All the authors contributed to the preparation of the manuscript. A.O. and M.J. supervised the research.

## Competing interests

A.O. and J.L. are co-inventors of a pending patent application on the presented method. The remaining authors declare no competing interests.
