## [Peer Review File · Nature Communications]

Rapid Sensing of Hidden Objects and Defects using a Single-Pixel Diffractive Terahertz SensorReviewers' Comments:

Reviewer #1 (Remarks to the Author):

The authors of the submitted manuscript propose a system that combines passive diffractive optics with THz-TDS and deep learning in the terahertz frequency region to determine the presence of defects by using only a single pixel. Their system successfully detects with sufficient probability the presence or absence of hidden subwavelength defects in the silicon substrate that are not visible from the outside.

Until now, single-pixel imaging has been performed by actively modulating the pattern of incident light using a spatial phase modulator or the like. In contrast, this research is new in that it focuses on the fact that even passive elements can discriminate image information in the terahertz frequency range by taking advantage of the fact that THz-TDS can obtain frequency information even in single pixels. The experiment was performed correctly, and the description of the methodology is considered adequate.

However, I am not convinced of the significance of this study. This is because it is much easier and more reliable to directly observe defects with a terahertz camera if the purpose is to detect the presence or absence of hidden defects. The method in this study requires training for deep learning in advance, and diffractive optics must be fabricated accordingly, which requires a great deal of advance preparation. It also requires significant effort to construct the THz-TDS system. On the other hand, with a terahertz camera, all that is needed is an appropriate terahertz light source. It is much easier to set up, and it provides information not only on the presence or absence of defects, but also on the location of defects. Unlike in the past, terahertz cameras are commercially available now and are less expensive than THz-TDS system.

Considering the above, I am unclear as to what the technical superiority of this method is. Since the method is unprecedented, it should be published in some journal, but I cannot think that Nature Communications would be appropriate.

Reviewer #2 (Remarks to the Author):

Jingxi Li et al. present an experimental study of rapid sensing of hidden defects by using a single-pixel Terahertz-TDS system. Authors demonstrate THz spectral shaping by multiple passive diffractive layers both before and after the sample materials. They relate the detection of defects/empty gaps to the relative ratio between two spectral peaks originating from the passive diffractive layers at 0.27 THz, and 0.37 THz, respectively. The authors show that with proper machine learning training procedures, voids inside silicon wafers can be effectively detected by this proposed technique. The results are interesting in terms of demonstrating one potentially viable solution for rapid terahertz imaging. However, I believe that the main results of rapid sensing by using diffractive terahertz spectroscopy are mostly incremental. What's more, the general effectiveness of this demonstrated technique is also not supported strongly enough by the results presented in the manuscript. I would like to add some comments on the core experimental results for defect detection inside the test samples by their diffractive THz processor.

- In Fig. 5e, there are significant fluctuations in many of the measured THz spectra. Such fluctuations, especially in some cases (e.g., sample no. 2 and no.9), would strongly couple into the effectiveness of defect detection. The authors should present the score marks in Fig.5c with error bars based on the data fluctuation in Fig. 5e. Then results from both sample 2 and sample 9 will partially break the threshold line of $S_{th} = 0.5$.
- Besides, the air gaps in silicon should be considered one of the easiest defects for detection. Even though the depth is relatively small (0.25mm), the lateral sizes of the gaps are large in most of the samples, 3 or 5 mm. The demonstrated performance on these easy samples is not very impressive. So, the general effectiveness of this proposed technique is questionable.
- Some important parameter details are missing. For instance, what is the THz beam profile distribution within the detection FOV? Furthermore, since mostly frequency components around 0.27 and 0.37 THz are relevant for defect detection, what are the beam profile distributions of these two components? Does this influence the detection effectiveness for defects located in different positions?

Reviewer #3 (Remarks to the Author):

Review of the paper draft

“Rapid Sensing of Hidden Objects and Defects using a Single-Pixel Diffractive Terahertz Processor”

I read the draft with interest. The paper investigates the use of diffraction mask as diffractive-combinatory elements within the application of object recognition at the terahertz wavelength. The specific embodiment is based on active illumination as it exploits a broadband source at terahertz radiation pre-processed with two diffractive layers and then a single-pixel time-domain spectroscopy detection pre-processed by the other two layers.

In the author’s proposed paradigm, the diffractive layers are determined via a machine learning approach in order to provide the detection of hidden objects via spectral analysis of the output. The paper reflects the current production of the exploitation of diffractive elements of the authors.

In summary, I found the paper very interesting, but in its current form, I also found some fundamental points unclear.

1) I find the specific definition of “Terahertz Processor” a bit overhyping. The device implements a static linear transformation (space-frequency) and while I think indeed a sensor I don’t think it embeds the inherent quality of a processor of information

2) I would invite the author to clarify the methodological innovation in the paper compared to ref 22-24. I certainly appreciate that a key advantage of the terahertz implementation stands within the scale of the diffraction layer used which allows a significantly simpler realization approach.

3) From the abstract I found the general statement “ As a proof-of-concept, we trained a diffractive terahertz processor to sense hidden defects (including subwavelength features) inside test samples, and evaluated its performance by analyzing the detection sensitivity as a function of the size and position of the unknown defects.” (and the many related passages I the paper) a bit misdirecting.

The proposed embodiment is not a near-field implementation and deals with a diffraction-limited spatial spectrum. In general, detecting scattering of a sub-wavelength point (which can be done even in diffraction-limited setting) is not equivalent to detecting a subwavelength feature.

While the concept of resolution limit is certainly more convoluted in presence of phase information that can be processed by a system, my understanding is that all the layers used are very far from the object (several wavelengths) and the detection is fundamentally based on the spectral intensity only. So, I would expect the morphological sensitivity of the system to be limited to super-wavelength features for any training (I stand corrected if I missed something in this specific aspect). I do appreciate that the diffraction limit within the silicon layer might be smaller because of the large index.

2) In addition, while I found the result very interesting, I would propose that the author should consider in their introduction that for the specific proof-of-concept proposed, (a defect in homogeneous material) reflective TDS is widely exploited to detect imperfections without imaging (with the additional advantage that time-of-flight is widely used to isolate the specific plane of the defect). I do appreciate, that the layers can be potentially programmed to recognise complicated morphologies, however, the narrative should convey a comparable case of use.

3) In Terms of motivation and background, I would highlight that the literature introduced in the background mostly refers to time-domain imaging approaches which tend to be slow because of the need of retaining time-domain information. On this specific aspect, the authors offer a comparison with an incoherent imaging approach

“In addition to these, currently available terahertz focal-plane arrays based on field-effect transistors and microbolometers do not provide time-resolved and frequency-resolved image data, limiting the types of structural information that can be detected 19,20. Due to these limitations, the space-bandwidth products (SBPs) of existing terahertz imaging systems are orders of magnitude lower than their counterparts operating in the visible band, limiting the overall system throughput”.

I find this passage quite not rigorous (i) the approach proposed in the paper uses the spectral axis as output and seems exploit the intensity spectrum, hence it does not exploit time-resolved sensing. The system seems completely linear, hence I would assume that a spectrally incoherent terahertz sources possessing the two specific wavelength components used (or more) would produce similar results. The vision proposed by the authors that an incoherent imaging array cannot be used for multispectral (incoherent) detection is possibly domain specific, as both in optics at terahertz tuneable sources and filters can be used.

(ii) In addition, it is not clear what space-bandwidth products (SBPs) refers to in this comparison. The approach proposed from the authors does not seem at this stage sensitive to the spectral information (in the sense that the frequency axis is not used to represent the frequency response in this specific scenario)

(iii) I suggest a refinement of the background literature:

-The passage

“... rich spectral signatures of different materials in the terahertz band” references ref 1-5 which appears a bit a random group. Ref 3,4,5 seems very recent while the topic is quite mature.

I could offer some relevant work to the author attention to credit some seminal work in the topic. Markelz et al. Chem. Phys. Lett 320 42 (2000).

Woolard et al. Phys. Rev. E. 65 051903 (2002).
Bolivar et al. Phys. Med. Biol. 47 3815 (2002).
Leahy-Hoppa et al. Chem. Phys. Lett. 434 227 (2007).
Schulkin & Zhang Laser Focus World 42 89 (2006).
Zeitler et al. J. Pharm. Pharmacol. 59 209 (2007).
Pickwell & Wallace J. Phys. D 39 R301 (2006).

-Regarding

“...can be resolved using computational methods without raster scanning.”

I would suggest that

Ref 17 should probably be Stantchev et al Sci. Adv. 2(6), e1600190 (2016) (which largely predates the chosen one).

Some minor comments

-In figure 1a the little bar with 10ps seems composed as a scale indication, while it is probably the full duration of the waveform.

-silicon has a significantly high refractive index, which makes the scattering cross-section of any defects quite large. Is that a requirement for the given SNR of the TDS system?

-I would probably comment that a diffraction limit within the silicon (index 3.418) is lower than the one in free space.

-I would also offer to the author that the proposed comparison with computational imaging approaches are used for their relatively high SNR compared to the standard raster scan (The author could refer to the large terahertz ghost imaging literature). While it is accurate that they are not as fast as a single shot detection, they are quite faster than raster scan approaches, and some embodiments enable accurate reconstruction of the near field.

The original referee comments are shown in black color, whereas for ease of communication, our answers are provided in blue. The changes made in the manuscript's main text and the Supplementary Information have been marked in yellow.

Reviewer #1 (Remarks to the Author):

The authors of the submitted manuscript propose a system that combines passive diffractive optics with THz-TDS and deep learning in the terahertz frequency region to determine the presence of defects by using only a single pixel. Their system successfully detects with sufficient probability the presence or absence of hidden subwavelength defects in the silicon substrate that are not visible from the outside.

Until now, single-pixel imaging has been performed by actively modulating the pattern of incident light using a spatial phase modulator or the like. In contrast, this research is new in that it focuses on the fact that even passive elements can discriminate image information in the terahertz frequency range by taking advantage of the fact that THz-TDS can obtain frequency information even in single pixels. The experiment was performed correctly, and the description of the methodology is considered adequate.

-- We sincerely thank the reviewer for his/her positive and constructive evaluations.

However, I am not convinced of the significance of this study. This is because it is much easier and more reliable to directly observe defects with a terahertz camera if the purpose is to detect the presence or absence of hidden defects. The method in this study requires training for deep learning in advance, and diffractive optics must be fabricated accordingly, which requires a great deal of advance preparation. It also requires significant effort to construct the THz-TDS system. On the other hand, with a terahertz camera, all that is needed is an appropriate terahertz light source. It is much easier to set up, and it provides information not only on the presence or absence of defects, but also on the location of defects. Unlike in the past, terahertz cameras are commercially available now and are less expensive than THz-TDS system. Considering the above, I am unclear as to what the technical superiority of this method is. Since the method is unprecedented, it should be published in some journal, but I cannot think that Nature Communications would be appropriate.

-- We thank the reviewer for these important comments. To address these concerns, we would like to first provide some additional clarification on the novelty and unique advantages presented by our reported defect detection framework. For this, we have added a new paragraph into the Discussion section, main text, which is quoted below:

"...We would like to further underscore some of the unique aspects of our reported framework over conventional approaches that utilize terahertz focal plane arrays or reflective TDS-based machine vision systems. Let us consider a common scenario in industrial quality control systems or security applications: hidden defects are discovered in only a minute fraction of the inspected objects, for example, just one in every 1000 or 10000 objects. This means that, in more than 99.9% of the inspected cases, the objects encountered are ordinary and defect-free.

However, for every inspected object, traditional terahertz imaging systems must undergo the following steps, one by one: (1) recording and reconstructing 3D images of each of these >1000-10000 objects using a focal plane array and/or a scanning beam system, (2) sequentially storing these 3D images and uploading them to a remote or local server, and then (3) running machine learning algorithms using GPU clusters to digitally determine the presence of hidden defects. Consequently, a substantial amount of imaging and computational resources are wasted in extracting and processing redundant information, leading to lowered detection throughput and a massive burden on data/storage and bandwidth.

In contrast, our method effectively circumvents all these steps. It does not require reconstructing or creating 3D images for each object, nor does it need to store and upload any images to the cloud, and it also certainly does not necessitate using GPU clusters to handle large volumes of data. Our method signifies a novel paradigm, relieving us entirely from the dependence on the image reconstruction, storage, and transmission steps needed by GPU-based digital processing systems. As a result, the efficiency of terahertz defect inspection/detection can be dramatically enhanced, and concurrently the cost per inspection can be substantially reduced using the presented diffractive defect sensors.

Regarding the reviewer's concern about the requirement of training in our framework, we added the following sentences in the Introduction and Results section of our revised manuscript:

*"... Once their deep learning-based training is complete (**which is a one-time effort**), the resulting diffractive layers are physically fabricated using 3D printing or additive manufacturing, which forms an optical neural network.³¹⁻⁴⁴ ..."*

In the Results section, the main text: ***"...Note that our data-driven training process is a one-time effort, similar to the training effort that a digital defect analyzer based on a THz camera would need to go through (using e.g., supervised learning) in an industrial or security setting. ..."***

Moreover, we would also like to clarify the reason for using a THz-TDS setup in our framework. For this purpose, we have added the following sentences to the Discussion section of our revised manuscript:

"In the demonstrated diffractive defect sensor, we utilized only two wavelength components at the output power spectra of the single-pixel detector to encode the defect information, and did not leverage the entire spectral bandwidth provided by the THz-TDS system. Our decision to use the THz-TDS system was primarily driven by the availability of hardware resources in our laboratory. Therefore, by using a THz illumination set-up with two pre-determined wavelengths (λ_1 and λ_2), the complexity of our current system would be significantly simplified, with costs substantially reduced. ..."

Reviewer #2 (Remarks to the Author):

Jingxi Li et al. present an experimental study of rapid sensing of hidden defects by using a single-pixel Terahertz-TDS system. Authors demonstrate THz spectral shaping by multiple passive diffractive layers both before and after the sample materials. They relate the detection of defects/empty gaps to the relative ratio between two spectral peaks originating from the passive diffractive layers at 0.27 THz,

and 0.37 THz, respectively. The authors show that with proper machine learning training procedures, voids inside silicon wafers can be effectively detected by this proposed technique. The results are interesting in terms of demonstrating one potentially viable solution for rapid terahertz imaging.

-- We sincerely thank the reviewer for his/her positive and constructive evaluations.

However, I believe that the main results of rapid sensing by using diffractive terahertz spectroscopy are mostly incremental. What's more, the general effectiveness of this demonstrated technique is also not supported strongly enough by the results presented in the manuscript.

I would like to add some comments on the core experimental results for defect detection inside the test samples by their diffractive THz processor.

- In Fig. 5e, there are significant fluctuations in many of the measured THz spectra. Such fluctuations, especially in some cases (e.g., sample no. 2 and no. 9), would strongly couple into the effectiveness of defect detection. The authors should present the score marks in Fig. 5c with error bars based on the data fluctuation in Fig. 5e. Then results from both sample 2 and sample 9 will partially break the threshold line of $S_{th} = 0.5$.

-- We thank the reviewer for this important comment. We want to emphasize that, as mentioned in our manuscript, we employed an averaging strategy during the detection process to mitigate SNR issues. To reflect this point, we have modified the following sentences in the Results section, main text:

“... To improve our SNR and build experimental resilience for hidden defect detection, we averaged the 5 spectral measurements for each test object and used the two peak spectral values at the resulting average spectrum for $s(\lambda_1)$ and $s(\lambda_2)$; see Fig. 5c. Stated differently, these 5 measurements collectively contributed to the representation of a single data point (i.e., the final detection score $\hat{s}_{exp}(\lambda)$) in Fig. 5c. ...”

Furthermore, we have conducted a new analysis on the impact of this averaging factor on the detection performance of our framework. For this purpose, we have created a new **Figure 6** and added a new paragraph into the Discussion section, main text, as quoted below:

*“...In our experimental results reported earlier using our single-pixel diffractive defect sensor, we adopted a strategy of taking the average of $N_{avg} = 5$ measurements to improve the detection SNR and the defect detection performance of the system. To delve deeper into this averaging strategy, similar to Fig. 5d, from the 10 spectral measurements of this defect-free sample, we randomly selected N_{avg} samples to form a group for averaging, and accordingly took the spectral peaks in the resulting spectrum as $s(\lambda_1)$ and $s(\lambda_2)$ to obtain the detection score s_{det} . After analyzing all the detection results corresponding to all the possible combinations, i.e., $C\left(\begin{smallmatrix} 10 \\ N_{avg} \end{smallmatrix}\right)$, we calculated the false positive rate, FPR, as a function of the averaging factor N_{avg} . **Figure 6** reports the impact of N_{avg} on the FPR in our detection results. These findings indicate that when N_{avg} is 1, i.e., no averaging is used, the FPR reaches as high as ~30%. **However, as N_{avg} increases, the reported FPR begins to decrease significantly. Specifically, when $N_{avg} = 5$, the FPR falls to 10.7%, which aligns with our earlier results in Fig. 5d. When N_{avg} further increases to 8, the FPR drops to 0%, indicating that false positives can be eliminated by this averaging strategy (see Fig. 6).”***

- Besides, the air gaps in silicon should be considered one of the easiest defects for detection. Even though the depth is relatively small (0.25 mm), the lateral sizes of the gaps are large in most of the samples, 3 or 5 mm. The demonstrated performance on these easy samples is not very impressive. So, the general effectiveness of this proposed technique is questionable.

-- To address the reviewer's concern about the size of the hidden defects in the test samples used for our experimental validation, we have added the following sentences into the Results section, main text:

*"... Note that in our experiments, the smallest defects we used have a size of $1 \times 3 \text{ mm}^2$ ($3 \times 1 \text{ mm}^2$), and such defects in samples No. 4, 6, and 7 were all located near the edges of the detection FOV, presenting particularly challenging detection cases. **This small defect area of 3 mm^2 approaches the smallest hidden feature that our diffractive sensor can detect as characterized in our simulation analysis ($1.25 \text{ mm} \times 1.25 \text{ mm} \approx 1.56 \text{ mm}^2$). Therefore, these experimental results constitute compelling evidence demonstrating the practical feasibility of our diffractive defect sensor.**"*

- Some important parameter details are missing. For instance, what is the THz beam profile distribution within the detection FOV? Furthermore, since mostly frequency components around 0.27 and 0.37 THz are relevant for defect detection, what are the beam profile distributions of these two components? Does this influence the detection effectiveness for defects located in different positions?

-- We thank the reviewer for these valuable questions. To answer these questions, we have created a new **Supplementary Fig. S2** to visualize the distribution of the THz beam profile within the detection field of view (FOV) and analyzed the impact of this profile on the defect detection results. We have added a new paragraph into the Discussion section, main text to report the details and conclusions for this analysis, as quoted below:

*"..Another aspect we would like to discuss pertains to the variation of our diffractive defect sensor's performance at different sub-regions within the detection FOV. For this, we simulated the terahertz wave field (E_{detFOV}) within the detection FOV at plane D, i.e., the plane where the incoming terahertz field interacts with the potential hidden defects (Supplementary Fig. 2a). The amplitude and phase distributions of E_{detFOV} at the two predetermined wavelengths ($\lambda_1 = 0.8 \text{ mm}$ and $\lambda_2 = 1.1 \text{ mm}$) are shown in Supplementary Fig. 2b-e. Interestingly, $E_{detFOV}(\lambda_1)$ exhibits a relatively uniform amplitude and phase distributions at the center, with the phase profile beginning to present faster changes as moving away from the center. On the other hand, the amplitude and phase distributions of $E_{detFOV}(\lambda_2)$ present a specific structure, featuring higher intensity in the center and lower intensity around. **These observations suggest that the terahertz light reaching the detection FOV, after being processed by the encoding diffractive network, primarily forms structured light illumination at λ_2 , while it largely maintains a spherical wave-like illumination pattern at λ_1 .** We further analyzed the influence of this structured light illumination on the defect detection results. As illustrated in Supplementary Fig. 2f, we assumed a hidden defect with $D_x = D_y = 3 \text{ mm}$, $D_z = 0.25 \text{ mm}$ that can be located anywhere within the detection FOV, identical to the one in Sample No. 1 used in our experimental validation. We numerically quantified the output spectral intensity at the two operational wavelengths, $s(\lambda_1)$ and $s(\lambda_2)$, as a function of the hidden defect's position within the detection FOV, and calculated the resulting final detection score s_{det} in each case. These results are summarized in Supplementary Fig. 2g-i. **As shown in Supplementary Fig. 2j, the detection scores s_{det} within the***

*detection FOV exceed the detection threshold $s_{th} = 0.5$, indicating 100% true positive detection, except at the edges of the detection FOV, which have relatively poor representation during the training phase. It can be seen that, $s(\lambda_1)$ presents a fairly uniform distribution (excluding the edges); in contrast, the overall distribution of $s(\lambda_2)$ shows a significant negative correlation with s_{det} , and they both exhibit substantial correspondence with the amplitude distribution of $E_{detFOV}(\lambda_2)$ as shown in Supplementary Fig. 2d. **This finding suggests that $E_{detFOV}(\lambda_1)$ and $E_{detFOV}(\lambda_2)$ essentially play roles akin to “reference” light and “probe” light, respectively.** Here, the intensity structure of this “probe” light $E_{detFOV}(\lambda_2)$ notably contributes to the distribution of s_{det} , which both exhibit features of higher values at the center and lower values at the periphery, correlating with our observations in Fig. 3j.”*

Reviewer #3 (Remarks to the Author):

I read the draft with interest. The paper investigates the use of diffraction mask as diffractive-combinatory elements within the application of object recognition at the terahertz wavelength. The specific embodiment is based on active illumination as it exploits a broadband source at terahertz radiation pre-processed with two diffractive layers and then a single-pixel time-domain spectroscopy detection pre-processed by the other two layers.

In the author’s proposed paradigm, the diffractive layers are determined via a machine learning approach in order to provide the detection of hidden objects via spectral analysis of the output. The paper reflects the current production of the exploitation of diffractive elements of the authors. In summary, I found the paper very interesting, but in its current form, I also found some fundamental points unclear.

-- We sincerely thank the reviewer for his/her positive and constructive evaluations.

1) I find the specific definition of “Terahertz Processor” a bit overhyping. The device implements a static linear transformation (space-frequency) and while I think indeed a sensor I don’t think it embeds the inherent quality of a processor of information

-- We thank the reviewer for this valuable comment. We have accordingly revised our manuscript by replacing the word “processor” with the word “sensor”. The others were also modified based on the context, as yellow highlighted in our revised manuscript files.

2) I would invite the author to clarify the methodological innovation in the paper compared to ref 22-24. I certainly appreciate that a key advantage of the terahertz implementation stands within the scale of the diffraction layer used which allows a significantly simpler realization approach.

-- We thank the reviewer for giving us an opportunity to clarify this point. All of these references refer to diffractive network-related works previously published by our group, but the methodological contribution of this manuscript has significant differences from these earlier works as detailed below.

In the original Ref. 22 (the new Ref. 32), we demonstrated the use of a diffractive optical network for broadband spectral operation (i.e., broadband diffractive network), where deterministic optical elements

such as tunable spectral filters were designed. However, this current work is on the design of encoder/decoder diffractive pairs to perform statistical inference on the input object (hidden defects), which presents a fundamental difference from the deterministic optical filters presented in the original Ref. 22 in terms of the operational principles, aims, training and system architecture design.

In the original Refs. 23-24 (the new Refs. 33-34), we focused on image classification tasks without any encoder/decoder pairs as used in our current manuscript. In these earlier works, we also introduced an optoelectronic hybrid design that cascades a diffractive optical network with an electronic neural network, which helps to boost image classification accuracy. **None of these are applicable or used in this work, and therefore there are fundamental differences between this current submission and our former publications in terms of the operational principles, aims/goals, training and system architecture design.**

To better highlight these unique points presented by the design in this work, we have modified the following sentences in our revised manuscript, in the Introduction section, main text:

“...Our reported approach represents the first demonstration that a diffractive optical network performs all-optical detection of hidden structures within 3D samples, which offers distinct advantages compared to the existing terahertz imaging and sensing systems used for the same purpose. ...”

In the Discussion section, main text:

“... This diffractive THz sensor features a distinctive architecture composed of a pair of encoder and decoder diffractive networks, each taking the unique responsibilities of structured illumination and spatial-spectral encoding, a configuration that has never been showcased in previous literature on diffractive networks. Based on this unique framework, we demonstrated a proof-of-concept hidden defect detection sensor. ...”

3) From the abstract I found the general statement “As a proof-of-concept, we trained a diffractive terahertz processor to sense hidden defects (including subwavelength features) inside test samples, and evaluated its performance by analyzing the detection sensitivity as a function of the size and position of the unknown defects.” (and the many related passages I the paper) a bit misdirecting.

The proposed embodiment is not a near-field implementation and deals with a diffraction-limited spatial spectrum. In general, detecting scattering of a sub-wavelength point (which can be done even in diffraction-limited setting) is not equivalent to detecting a subwavelength feature. While the concept of resolution limit is certainly more convoluted in presence of phase information that can be processed by a system, my understanding is that all the layers used are very far from the object (several wavelengths) and the detection is fundamentally based on the spectral intensity only. So, I would expect the morphological sensitivity of the system to be limited to super-wavelength features for any training (I stand corrected if I missed something in this specific aspect). I do appreciate that the diffraction limit within the silicon layer might be smaller because of the large index.

-- We thank the reviewer for these insightful comments. To better clarify these points, we have added the following sentences to the Results, main text:

“... It should be noted that subwavelength features used in our manuscript specifically refer to lateral features larger than the diffraction limit in air ($\sim 0.5\lambda_m$) but smaller than λ_m”

We have also brought this clarification into the Abstract and Introduction section of our manuscript:

(In the Abstract)

“... As a proof-of-concept, we trained a diffractive terahertz network to sense hidden defects (including diffraction-limited subwavelength features) inside test samples, and evaluated its performance by analyzing the detection sensitivity as a function of the size and position of the unknown defects. ...”

(In the Introduction section, main text)

“... We numerically analyzed the performance of our diffractive defect sensor by evaluating its detection sensitivity as a function of the size and the position of the hidden defects within the detection field-of-view (FOV), also covering subwavelength feature sizes that are within the diffraction limit of light. ...”

In addition, regarding the reviewer’s comment on the smaller diffraction limit within the silicon layer, we have added the following sentence to the Results, main text:

“... Despite the fact that the diffraction limit would be smaller in a high-refractive-index material like silicon, the medium between the sample under test and the detector is air, which sets an upper limit of 1 on the effective numerical aperture (NA) of the detection system. ...”

2) In addition, while I found the result very interesting, I would propose that the author should consider in their introduction that for the specific proof-of-concept proposed, (a defect in homogeneous material) reflective TDS is widely exploited to detect imperfections without imaging (with the additional advantage that time-of-flight is widely used to isolate the specific plane of the defect). I do appreciate, that the layers can be potentially programmed to recognise complicated morphologies, however, the narrative should convey a comparable case of use.

-- To address the reviewer’s comment, we have modified the following sentence in the Introduction section, main text to include the reflective THz-TDS into our context:

“... However, existing THz-TDS systems (including reflective versions) are single-pixel and require raster scanning to acquire the image of the hidden features, resulting in relatively low-speed/low-throughput systems. ...”

We have also provided some additional clarification on the novelty and unique advantages presented by our reported defect detection framework compared to other systems, including reflective THz-TDS-based approaches. For this purpose, we have added a new paragraph into the Discussion section, main text, which is quoted below:

“...We would like to further underscore some of the unique aspects of our reported framework over conventional approaches that utilize terahertz focal plane arrays or reflective TDS-based machine vision systems. Let us consider a common scenario in industrial quality control systems or security applications: hidden defects are discovered in only a minute fraction of the inspected objects, for example, just one in every 1000 or 10000 objects. This means that, in more than 99.9% of the inspected

cases, the objects encountered are ordinary and defect-free. However, for every inspected object, traditional terahertz imaging systems must undergo the following steps, one by one: (1) recording and reconstructing 3D images of each of these >1000-10000 objects using a focal plane array and/or a scanning beam system, (2) sequentially storing these 3D images and uploading them to a remote or local server, and then (3) running machine learning algorithms using GPU clusters to digitally determine the presence of hidden defects. Consequently, a substantial amount of imaging and computational resources are wasted in extracting and processing redundant information, leading to lowered detection throughput and a massive burden on data/storage and bandwidth.

In contrast, our method effectively circumvents all these steps. It does not require reconstructing or creating 3D images for each object, nor does it need to store and upload any images to the cloud, and it also certainly does not necessitate using GPU clusters to handle large volumes of data. Our method signifies a novel paradigm, relieving us entirely from the dependence on the image reconstruction, storage, and transmission steps needed by GPU-based digital processing systems. As a result, the efficiency of terahertz defect inspection/detection can be dramatically enhanced, and concurrently the cost per inspection can be substantially reduced using the presented diffractive defect sensors.

We have also added the following sentence into the Discussion section, main text:

“... Compared to conventional reflective THz-TDS defect detection systems used for similar purposes, which necessitate mechanical scanning to detect defects at various locations, this reflective diffractive sensor design would allow for all-optical, rapid defect detection within a large sample volume, eliminating the need for mechanical scanning or extensive data storage, transmission and digital processing.”

Moreover, we would also like to clarify the reason for using a THz-TDS setup in our framework. For this purpose, we have added the following sentences to the Discussion section of our revised manuscript:

“...In the demonstrated diffractive defect sensor, we utilized only two wavelength components at the output power spectra of the single-pixel detector to encode the defect information, and did not leverage the entire spectral bandwidth provided by the THz-TDS system. Our decision to use the THz-TDS system was primarily driven by the availability of hardware resources in our laboratory. Therefore, by using a THz illumination set-up with two pre-determined wavelengths (λ_1 and λ_2), the complexity of our current system would be significantly simplified, with costs substantially reduced. ...”

3) In Terms of motivation and background, I would highlight that the literature introduced in the background mostly refers to time-domain imaging approaches which tend to be slow because of the need of retaining time-domain information. On this specific aspect, the authors offer a comparison with an incoherent imaging approach

“In addition to these, currently available terahertz focal-plane arrays based on field-effect transistors and microbolometers do not provide time-resolved and frequency-resolved image data, limiting the types of structural information that can be detected^{19,20}. Due to these limitations, the space-bandwidth

products (SBPs) of existing terahertz imaging systems are orders of magnitude lower than their counterparts operating in the visible band, limiting the overall system throughput”.

I find this passage quite not rigorous

(i) the approach proposed in the paper uses the spectral axis as output and seems exploit the intensity spectrum, hence it does not exploit time-resolved sensing. The system seems completely linear, hence I would assume that a spectrally incoherent terahertz sources possessing the two specific wavelength components used (or more) would produce similar results. The vision proposed by the authors that an incoherent imaging array cannot be used for multispectral (incoherent) detection is possibly domain specific, as both in optics at terahertz tuneable sources and filters can be used.

-- Throughout our manuscript, when discussing coherence, we merely asserted that "**we treat the detection system as a coherent diffractive processor.**" **It is important to note that the coherence referred to here pertains to spatial coherence, not temporal coherence.**

In fact, in our presented system, the input illumination is a terahertz pulse that spans a broad continuous spectrum within the terahertz band, hence devoid of temporal coherence.

Therefore, it's true that a spectrally incoherent terahertz source with two pre-determined wavelengths can be used to produce similar results. Moreover, we have never indicated in the text that a (temporally) incoherent sensor array cannot be used for multi-wavelength detection. The single-pixel spectral detector we utilized does not require any coherence conditions as a prerequisite; only the diffractive network system necessitates a spatial coherence condition. To emphasize these points in our revised manuscript, we have added the following sentences into the Results section, main text:

“... The forward model of this design can be treated as a coherent optical system that processes spatially coherent terahertz waves at a predetermined set of 2 wavelengths (λ_1 and λ_2), where the resulting diffraction and interference processes are used for the defect detection task. As depicted in Fig. 1a, a set of diffractive layers is positioned before the sample under test to provide spatially coherent, structured broadband illumination within a given detection FOV, acting as an all-optical front-end network that is trainable. ...”

Moreover, we have added the following sentences to the Discussion section of our revised manuscript:

*“...In the demonstrated diffractive defect sensor, we utilized only two wavelength components at the output power spectra of the single-pixel detector to encode the defect information, and did not leverage the entire spectral bandwidth provided by the THz-TDS system. Our decision to use the THz-TDS system was primarily driven by the availability of hardware resources in our laboratory. **Therefore, by using a THz illumination set-up with two pre-determined wavelengths (λ_1 and λ_2), the complexity of our current system would be significantly simplified, with costs substantially reduced.** ...”*

(ii) In addition, it is not clear what space-bandwidth products (SBPs) refers to in this comparison. The approach proposed from the authors does not seem at this stage sensitive to the spectral information (in the sense that the frequency axis is not used to represent the frequency response in this specific scenario)

-- We would like to clarify that space-bandwidth product (SBP) is the product of the FOV (space) and the spatial frequency range (bandwidth), where the bandwidth stands for **the spatial frequency bandwidth, not spectral bandwidth**.

Since SBP measures the information throughput that an optical imaging system can provide, this specific sentence we stated in the manuscript means that terahertz imaging systems can provide significantly less information on the input object/scene compared to their counterparts operating in the visible bands, which is due to the limited pixel count and resolution achievable by existing THz imaging systems.

To emphasize this point in our revised manuscript, we have added the following sentences into the Introduction section, main text:

“... In addition to these, currently available terahertz focal-plane arrays based on field-effect transistors and microbolometers offer a limited spatial resolution and do not provide time-resolved and frequency-resolved image data, limiting the types of structural information that can be detected.^{29,30} Due to these limitations, the space-bandwidth products (SBPs) of existing terahertz imaging systems are orders of magnitude lower than their counterparts operating in the visible band, thereby constraining the system's overall information throughput and its capacity to adequately capture the desired details of the hidden structures of interest.”

(iii) I suggest a refinement of the background literature:

-The passage “... rich spectral signatures of different materials in the terahertz band” references ref 1-5 which appears a bit a random group. Ref 3,4,5 seems very recent while the topic is quite mature.

I could offer some relevant work to the author attention to credit some seminal work in the topic.

Markelz et al. Chem. Phys. Lett 320 42 (2000).

Woolard et al. Phys. Rev. E. 65 051903 (2002).

Bolivar et al. Phys. Med. Biol. 47 3815 (2002).

Leahy-Hoppa et al. Chem. Phys. Lett. 434 227 (2007).

Schulkin & Zhang Laser Focus World 42 89 (2006).

Zeitler et al. J. Pharm. Pharmacol. 59 209 (2007).

Pickwell & Wallace J. Phys. D 39 R301 (2006).

- Regarding “...can be resolved using computational methods without raster scanning.” I would suggest that Ref 17 should probably be Stantchev et al Sci. Adv. 2(6), e1600190 (2016) (which largely predates the chosen one).

-- We appreciate the reviewer's suggestions regarding related literature. We have updated the references accordingly; see **Refs. 6-12**. We have also replaced the original Ref. 17 with the older one suggested by the reviewer, which is now cited as **Ref. 24**.

Some minor comments

- In figure 1a the little bar with 10ps seems composed as a scale indication, while it is probably the full duration of the waveform.

-- For this clarification, we have added the following sentences to the captions of Figure 1:

“... The input illumination of the system shown here is provided by an ultrashort terahertz pulse, with an overall duration of ~320 ps. For illustrative purposes, only the segment with a significant magnitude is shown. ...”

- silicon has a significantly high refractive index, which makes the scattering cross-section of any defects quite large. Is that a requirement for the given SNR of the TDS system?

-- We would like to clarify that our use of silicon wafers here solely aims to demonstrate that the system can be utilized to detect defects hidden within silicon materials due to their broad use/applications in quality control and security settings. We have mentioned this point in the Results section, main text:

“To demonstrate the feasibility of our nondestructive diffractive defect detection framework, we designed a proof-of-concept single-pixel diffractive terahertz sensor that can effectively detect pore-like hidden defects within silicon materials; these defects are not visible from the outside. Such a capability is highly sought after in numerous industrial applications due to its high prevalence and significance in determining e.g., the quality, reliability, and performance of manufactured parts/products. ...”

- I would probably comment that a diffraction limit within the silicon (index 3.418) is lower than the one in free space.

-- To address the reviewer's comment here, we have added the following sentence into the Results, main text:

“... Despite the fact that the diffraction limit would be smaller in a high-refractive-index material like silicon, the medium between the sample under test and the detector is air, which sets an upper limit of 1 on the effective numerical aperture (NA) of the detection system. ...”

- I would also offer to the author that the proposed comparison with computational imaging approaches are used for their relatively high SNR compared to the standard raster scan (The author could refer to the large terahertz ghost imaging literature). While it is accurate that they are not as fast as a single shot detection, they are quite faster than raster scan approaches, and some embodiments enable accurate reconstruction of the near field.

-- To address the reviewer's comment regarding the THz computational imaging approaches including THz computational ghost imaging, we have incorporated more references to the related literature, including **Refs. 21, 23 and 25**. We have also added the following sentence into the Introduction section, main text:

“... This approach, often known as terahertz computational ghost imaging, can achieve high image SNR with a decent spatial resolution. ...”

Also, the downsides of some of these computational imaging approaches have been mentioned in the sentence following the one quoted above:

“... However, the physical constraints of spatial light modulators operating at terahertz wavelengths limit the speed, and increase the size, cost, and complexity of these imaging systems. ...”

To conclude, we sincerely thank the editor and reviewers for their constructive comments and feedback, which helped us to further improve the quality and clarity of our manuscript.

REVIEWER COMMENTS

Reviewer #1 (Remarks to the Author):

The manuscript has been well revised to answer my points and clarify its significance. I recommend the publication of this manuscript.

Reviewer #2 (Remarks to the Author):

I have carefully read all the reviewer comments and responses from the authors. I would not recommend publication of the revised manuscript in the journal of Nature communications, for the following two major considerations. First, despite the arguments from the authors in the response letter, the results presented in this manuscript should still be considered as mostly incremental compared to their earlier publications. Second, the significance of this study is not convincing from THz imaging point of view. In this regard, I share the same opinion as another reviewer that similar air gaps in silicon which were studied in this manuscript can be super easily and quickly spotted with inexpensive THz cameras without going through any complicated computer algorithms. Overall, this manuscript deserves publication in some journals, but I cannot recommend publication in Nature Communications.

Reviewer #3 (Remarks to the Author):

The author resubmitted an amended version. I re-iterate that the paper is very interesting. I think most modifications implemented by the authors are appropriate.

However, respectfully, I am but confused regarding the answers provided on the point raised about discerning subwavelength features.

The authors answered with the modifications

"... It should be noted that subwavelength features used in our manuscript specifically refer to lateral features larger than the diffraction limit in air ($\sim 0.5\lambda$) but smaller than λ"

which does not seem to address the point raised regarding the claim of being able to detect subwavelength features (i.e. morphological awareness of subwavelength defect) against the simple detection of a subwavelength defect simply because it certainly contains a super-wavelength spatial spectrum. Said differently, the system cannot process any near-field, so any sufficiently small defect would look morphologically indistinguishable regardless of the training.

In addition the modification

".....(including diffraction-limited subwavelength features)" sounds now logically incorrect as features cannot be "diffraction limited" (which is a property of an imaging system).

Similarly, I am not sure what the passage in response to the same point, now stating "... We numerically analyzed the performance of our diffractive defect sensor by evaluating its detection sensitivity as a function of the size and the position of the hidden defects within the detection field-of-view (FOV), also covering subwavelength feature sizes that are within the diffraction limit of light. ..."

Is actually supporting. From an imaging point of view, any localised subwavelength feature has trivially a wide scattered spatial spectrum that overlaps with the diffraction-limited fraction of the

spectrum acquired. This is normal and does not mean you are sensitive to sub-wavelength features. It just means that any sub-wavelength feature is blurred and indistinguishable from a super-wavelength defect that has the same scattered spatial spectral components in the diffraction-limited acquisition bandwidth.

In summary, while I think the authors can claim that their sensitivity is sufficient to detect smaller-than-wavelength defects, I think the paper should not imply the idea that the network could be trained to detect a sub-wavelength features (i.e. discriminate a sub-wavelength defect from another).

The original referee comments are shown in black color, whereas for ease of communication, our answers are provided in blue. The changes made in the manuscript's main text and the Supplementary Information have been marked in yellow.

Reviewer #1 (Remarks to the Author):

The manuscript has been well revised to answer my points and clarify its significance. I recommend the publication of this manuscript.

-- We sincerely thank the reviewer for his/her positive evaluations and constructive feedback provided during the review process.

Reviewer #2 (Remarks to the Author):

I have carefully read all the reviewer comments and responses from the authors. I would not recommend publication of the revised manuscript in the journal of Nature communications, for the following two major considerations. First, despite the arguments from the authors in the response letter, the results presented in this manuscript should still be considered as mostly incremental compared to their earlier publications. Second, the significance of this study is not convincing from THz imaging point of view. In this regard, I share the same opinion as another reviewer that similar air gaps in silicon which were studied in this manuscript can be super easily and quickly spotted with inexpensive THz cameras without going through any complicated computer algorithms. Overall, this manuscript deserves publication in some journals, but I cannot recommend publication in Nature Communications.

-- We thank the reviewer for his/her valuable comments. To begin with, regarding the novelty of this work, we would like to emphasize that the diffractive network design developed in this work has significant differences compared to the diffractive designs in our previous works. Specifically, the current design represents the first-ever demonstration of using deep learning-optimized diffractive materials to all-optically detect hidden structures within 3D samples. Moreover, it distinctively incorporates the joint optimization of an encoder (for generating structured illumination) and a decoder diffractive network (for performing space-to-spectrum transformation) in a single system to collaboratively perform a statistical inference task using a single-pixel detector; this has also never been demonstrated in any work before. To better reflect these points in our manuscript, we have added the following sentences in the Introduction section, quoted below:

"...Our reported approach represents the first demonstration of all-optical detection of hidden structures within 3D samples, enabled by a single-pixel spectroscopic terahertz detector, entirely eliminating the need to scan the samples or create, store and digitally process their images. Our design employs a novel optical architecture featuring a passive diffractive encoder that generates structured illumination impinging onto the 3D sample of interest, coupled with a diffractive decoder that performs space-to-spectrum transformation, achieving defect detection based on the optical fields scattered from the sample volume. Leveraging this synergy between the two diffractive networks and their joint training/optimization, this single-pixel defect processor offers distinct advantages compared to the

existing terahertz imaging and sensing systems used for the same purpose. First, the hidden defect detection is accomplished using a single-pixel spectroscopic detector, eliminating the need for a focal plane array or raster scanning, thus greatly simplifying and accelerating the defect detection process. Second, the diffractive layers we employ are passive optical components, enabling our diffractive defect sensor to analyze the test object volume without requiring any external power source except for the terahertz illumination and single-pixel detector. Third, our all-optical end-to-end detection process negates the need for memory, data/image transmission or digital processing using e.g., a graphics processing unit (GPU), resulting in a high-throughput defect detection scheme.”

In addition, when comparing the significance of our work to existing imaging-based defect detection methodologies, we would like to emphasize once again that the single-pixel, non-imaging nature of our design effectively negates the need for 2D/3D image capture and subsequent image data storage, transmission and digital processing. This distinction grants our approach pronounced advantages in terms of detection throughput, computational burden, system complexity and cost. To further highlight these advantages, we have incorporated the following sentences into the Introduction section of our revised manuscript, as quoted below:

“... Overall, these characteristics render our single-pixel diffractive terahertz sensors particularly well-suited for high-throughput screening applications such as in industrial settings, e.g., manufacturing and security. These applications require high-throughput defect detection, where the hidden defects or objects of interest are often rare, but critically important to catch. Unlike conventional imaging-based methods, which are often hindered by the 3D image data overload due to redundant information and limited frame rates of 2D image sensors, our non-imaging and single-pixel defect detection approach can deliver markedly higher sensing throughput while offering cost-effectiveness and simplicity.”

Furthermore, we have expanded the Discussion section of our revised manuscript, as quoted below:

“... Even for large defects that are relatively simple to detect using state-of-the-art algorithms frequently used in machine vision, the throughput of such imaging-based solutions would still be limited by the low frame rate of the 2D image sensor-arrays; this image capture process will be even slower for various point-scanning-based THz imaging systems. In contrast, our method effectively circumvents all these steps. It does not require reconstructing or creating 3D images for each object, nor does it need to store and upload any images to the cloud, and it also certainly does not necessitate using GPU clusters to handle large volumes of data. Using a single-pixel detector and snap-shot dual illumination wavelengths for defect sensing, our method signifies a novel paradigm, eliminating the image capture/reconstruction, storage, and transmission steps needed by GPU-based digital processing systems. As a result, the volumetric defect detection rate can be elevated to align with the exceptional speed of single-pixel sensors, which can have a response time of $< 1 \mu s$ ⁴⁹. ...”

In summary, we firmly believe that the single-pixel diffractive terahertz sensor scheme presented in this manuscript is notably innovative and possesses significant advantages in various application scenarios, setting it apart from previous works and existing imaging-based detection methods.

Reviewer #3 (Remarks to the Author):

The author resubmitted an amended version. I re-iterate that the paper is very interesting.

I think most modifications implemented by the authors are appropriate.

-- We sincerely thank the reviewer for his/her positive evaluations and constructive feedback provided during the review process.

However, respectfully, I am but confused regarding the answers provided on the point raised about discerning subwavelength features.

The authors answered with the modifications

“... It should be noted that subwavelength features used in our manuscript specifically refer to lateral features larger than the diffraction limit in air ($\sim 0.5\lambda_m$) but smaller than λ_m”

which does not seem to address the point raised regarding the claim of being able to detect subwavelength features (i.e. morphological awareness of subwavelength defect) against the simple detection of a subwavelength defect simply because it certainly contains a super-wavelength spatial spectrum. Said differently, the system cannot process any near-field, so any sufficiently small defect would look morphologically indistinguishable regardless of the training.

In addition the modification

“..... (including diffraction-limited subwavelength features)” sounds now logically incorrect as features cannot be “diffraction limited” (which is a property of an imaging system).

Similarly, I am not sure what the passage in response to the same point, now stating

“... We numerically analyzed the performance of our diffractive defect sensor by evaluating its detection sensitivity as a function of the size and the position of the hidden defects within the detection field-of-view (FOV), also covering subwavelength feature sizes that are within the diffraction limit of light. ...”

Is actually supporting. From an imaging point of view, any localised subwavelength feature has trivially a wide scattered spatial spectrum that overlaps with the diffraction-limited fraction of the spectrum acquired. This is normal and does not mean you are sensitive to sub-wavelength features. It just means that any sub-wavelength feature is blurred and indistinguishable from a super-wavelength defect that has the same scattered spatial spectral components in the diffraction-limited acquisition bandwidth.

In summary, while I think the authors can claim that their sensitivity is sufficient to detect smaller-than-wavelength defects, I think the paper should not imply the idea that the network could be trained to detect a sub-wavelength feature (i.e. discriminate a sub-wavelength defect from another).

-- We thank the reviewer for these valuable comments and giving us the opportunity to improve the clarity of our manuscript. To address these important concerns and provide better clarification for the readers, we have added the following new sentences in the Discussion section of our revised manuscript, quoted as follows:

*“...Finally, our framework can potentially sense the presence of even smaller hidden defects or objects with subwavelength dimensions. While our diffractive defect sensor is diffraction-limited, isolated subwavelength features/defects can still generate traveling waves (through scattering) to be sensed by our diffractive layers. This, however, does **not** mean our diffractive processor can resolve two closely positioned subwavelength defects or morphologically distinguish them from larger defects since it can only process propagating waves from the defect volume, without access to the evanescent waves in the near-field of a defect that carry the super-resolution information. In this respect, similar to localization-based microscopic imaging approaches, it is potentially feasible to sense isolated subwavelength defects within the sample volume without any structural fine details or the capability to distinguish them from larger defects. This would be possible only with sufficient detection sensitivity: if the weak scattering of such small defects, coupled into propagating secondary waves, can be detected within the SNR of the single-pixel defect detection system. ...”*

We have also modified the original sentence of:

*“... It should be noted that **subwavelength** features used in our manuscript specifically refer to lateral features larger than the diffraction limit in air ($\sim 0.5\lambda_m$) but smaller than λ_m”*

as:

*“... It should be noted that the **smallest** defect used in our analysis has a size close to the diffraction limit of light in air ($\sim 0.5\lambda_m$). ...”*

Also, the original sentence in the Abstract:

*“... we trained a diffractive terahertz network to sense hidden defects (including **diffraction-limited subwavelength** features) inside test samples, and evaluated its performance by analyzing the detection sensitivity as a function of the size and position of the unknown defects ...”*

has now been changed to:

“... we trained a diffractive terahertz network to sense hidden defects inside test samples, and evaluated its performance by analyzing the detection sensitivity as a function of the size and position of the unknown defects...”

To conclude, we sincerely thank the editor and reviewers for their constructive comments and feedback, which helped us to further improve the quality and clarity of our manuscript.

REVIEWERS' COMMENTS

Reviewer #2 (Remarks to the Author):

The authors have nicely addressed all my comments and concerns, and I would like to recommend the publication of this revised manuscript.

Reviewer #3 (Remarks to the Author):

I believe the paper to be acceptable for publication.